# Large differences in carbohydrate degradation and transport potential among lichen fungal symbionts

Philipp Resl [1,2], Adina R. Bujold [3], Gulnara Tagirdzhanova [3], Peter Meidl[2], Sandra Freire Rallo [4], Mieko Kono[5], Samantha Fernández-Brime [5], Hörður Guðmundsson [6], Ólafur Sigmar Andrésson [6], Lucia Muggia [7], Helmut Mayrhofer [1], John P. McCutcheon[8,9], Mats Wedin [5], Silke Werth[2], Lisa M. Willis [3] & Toby Spribille[3✉]

Lichen symbioses are thought to be stabilized by the transfer of fixed carbon from a photosynthesizing symbiont to a fungus. In other fungal symbioses, carbohydrate subsidies correlate with reductions in plant cell wall-degrading enzymes, but whether this is true of lichen fungal symbionts (LFSs) is unknown. Here, we predict genes encoding carbohydrate-active enzymes (CAZymes) and sugar transporters in 46 genomes from the *Lecanoromycetes*, the largest extant clade of LFSs. All LFSs possess a robust CAZyme arsenal including enzymes acting on cellulose and hemicellulose, confirmed by experimental assays. However, the number of genes and predicted functions of CAZymes vary widely, with some fungal symbionts possessing arsenals on par with well-known saprotrophic fungi. These results suggest that stable fungal association with a phototroph does not in itself result in fungal CAZyme loss, and lends support to long-standing hypotheses that some lichens may augment fixed $CO_2$ with carbon from external sources.

[1] University of Graz, Institute of Biology, Universitätsplatz 2, 8010 Graz, Austria. [2] Ludwig-Maximilians-University Munich, Faculty of Biology Department 1, Diversity and Evolution of Plants, Menzingerstraße 67, 80638 Munich, Germany. [3] University of Alberta, Biological Sciences CW405, Edmonton, AB T6G 2R3, Canada. [4] Rey Juan Carlos University, Departamento de Biología y Geología, Física y Química Inorgánica, Móstoles, Spain. [5] Swedish Museum of Natural History, Botany Department, PO Box 50007, SE10405 Stockholm, Sweden. [6] Faculty of Life and Environmental Sciences, University of Iceland, Sturlugata 7, 102 Reykjavík, Iceland. [7] University of Trieste, Department of Life Sciences, via L. Giorgieri 10, 34127 Trieste, Italy. [8] Division of Biological Sciences, University of Montana, Missoula, MT, USA. [9]Present address: Biodesign Institute and School of Life Sciences, Arizona State University, Tempe, AZ, USA. ✉email: toby.spribille@ualberta.ca

Stable fungal associations with algae and/or cyanobacteria, usually referred to as lichens, feature prominently in the history of the discovery and study of symbiosis. In describing the pairing of fungi with microbial photosymbionts for the first time, the Swiss botanist Simon Schwendener proposed that lichen fungal symbionts derive nutrition from "assimilates" of their photosynthesizing partners[1]. Almost a hundred years later, Smith and colleagues revealed these transferred photosynthates to be polyols and glucose in the case of algae and cyanobacteria, respectively[2]. They and others traced the transfer of algal fixed carbon into fungal cells, where they found it to be converted into mannitol and arabitol[3,4]. The fungal-photosymbiont relationship is widely interpreted as conferring net independence from external carbohydrates on the resulting lichen thallus. Accordingly, lichen fungal symbionts have been classified as biotrophs[5], and the symbiotic outcome, the lichen thallus, as a "photosynthetic carbon autotroph"[6] or "composite autotroph"[7].

Fungi are assimilative heterotrophs and thus require a robust machinery of enzymes for scavenging and transporting extracellular nutrients, including carbohydrates. In arbuscular mycorrhizal and ectomycorrhizal fungi, the stable supply of glucose from plants is thought to have led to erosion or loss in many families of carbohydrate-active enzymes (CAZymes)[8,9], reflecting a common pattern of compensated trait loss in symbioses[10]. So what happened to CAZymes in lichen fungal symbionts? Multiple lines of indirect evidence have emerged over the last 40 years to suggest the retention of diverse CAZymes in lichen fungal symbionts (LFSs) or their secondary evolutionary derivates, especially specific plant cell wall-degrading enzymes (PCWDEs). First, molecular phylogenetic studies have shown multiple independent origins of putative saprotroph lineages from lichen fungal symbiont ancestors, both ancient[11] and recent[12–14]. How these newly evolved lineages acquired the carbohydrate breakdown arsenal they would presumably need for life without an alga has not been explained. Second, some fungi near the symbiont-to-saprotroph transition have been shown to switch between the two lifestyles facultatively, so-called "optional lichens"[15,16]; in these cases, the fungus appears not to be obligately dependent on the alga for nutrition. Third, many lichen symbioses exhibit anomalous "substrate specificity", i.e., they are restricted to specific organic substrates and unable to colonize others, suggesting lack of nutritional autonomy[17–19]. Fourth, lichen fungi are capable of growing axenically in vitro on a variety of sugars other than sugar alcohols, including crystalline cellulose and sucrose (reviewed by Fahselt[20]). Finally, enzymes involved in breakdown of lichen-exogenous polymers, including cellulose and lignin, have been isolated from lichens in nature (reviewed by Beckett et al.[21]). These phenomena could be dismissed as exceptions, but their distribution across the fungal symbiont tree hints at deeper underlying fungal capabilities, which if combined with phototroph symbiosis could lead to a kind of "hybrid" lifestyle, in which carbohydrates are obtained from multiple sources. The possibility of multiple carbon sources for lichens was even suggested by Schwendener himself in his original 1869 paper, in which he predicted that two tracks of nutrient acquisition would ultimately be proven: one for lichens that have minimal substrate contact, which he predicted to depend mostly on algal assimilates; and one for lichens that closely hug organic substrates such as tree bark or wood[1].

Unlike for many fungi, phenotypic carbohydrate use profiles have seldom, if ever, been developed for LFSs. This is in part due to the recalcitrance of most LFSs to culturing and their extreme slow growth, if culturing is successful. Knowledge gaps around unculturable or slow-growing fungi are common, but have been offset in recent years by genome sequencing. Coupled with the development of widely available databases such as CAZy[22], it has become possible to infer CAZymes for species for which a genome, but no experimental evidence, is currently available. Comparative genomic overviews of CAZyme repertoires are now available for many symbiotic fungi[9], but no survey exists of comparative CAZyme arsenals in LFSs.

Given the common assumption that lichen symbiont complementarity confers collective autotrophy on the symbiosis, and past results inferring they evolved from saprotrophic or non-lichen biotrophic ancestors (reviewed by Spribille et al.[23]), we hypothesized that LFSs would exhibit functional losses in CAZymes coinciding with the beginning of stable association with phototroph symbionts, similar to what has been found for PCWDEs in arbuscular- and ectomycorrhizal fungi[9,24,25]. To test this, however, we would need to map CAZymes across a well-sampled phylogeny and reconstruct the ancestral states of common ancestors, many of which are considerably older than the reconstructed origins of e.g., ectomycorrhizal fungi[26]. Here we map the occurrence of genes encoding CAZymes at two levels: across representative species of the Leotiomyceta group of Ascomycota, including the origin of the Lecanoromycetes, the largest extant lineage of LFSs; and among major groups within Lecanoromycetes, including species representing different ecological substrate specificities (specialists and generalists) as well as major morphological outcomes of the lichens they occur in (crusts, macrolichens). Our survey of 46 lecanoromycete genomes, 29 of which we sequenced for this project, reveals a complex pattern of retention and loss that lends support to Schwendener's hypothesis of hidden saprotrophy in some lichens and is not unequivocally consistent with CAZyme erosion upon acquisition of phototroph symbionts.

## Results

**Data set and phylogenomic reconstruction**. We assembled a data set of 83 fungal genomes, including 46 from the class Lecanoromycetes (Supplementary Table 1). Our sampling of Ascomycota genomes outside of Lecanoromycetes was informed by two considerations: 1) selected genomes should be representative of a range of CAZyme repertoires, already mapped by Miyauchi et al.[9], amongst others; and 2) they should draw primarily from Leotiomyceta, the group that includes the sibling classes of Lecanoromycetes. Because the few published lecanoromycete genomes currently over-represent fungal symbionts of generalist lichens in one taxonomic group within the subclass Lecanoromycetidae—the order Lecanorales—we generated 29 new genomes for this study, with emphasis on capturing both a diversity of substratum specializations as well as diverse lineages of the subclass Ostropomycetidae (sampling explained in greater detail in Methods). Eighteen of the genomes were obtained as metagenome-assembled genomes (MAGs; Supplementary Table 2). Completeness and quality metrics were comparable for genomes derived from culture and MAGs (Supplementary Figs. 1 and 2). Phylogenomic analysis based on 1310 inferred universally present single-copy orthologs (Fig. 1; Supplementary Fig. 3 and 4 and Supplementary Table 7) recovered major clades and sister group relationships found in recent studies, both among class-level clades of sampled Ascomycota[27] as well as within the Lecanoromycetes[28,29]. For each genome, we performed ab initio gene predictions and obtained functional annotations (CAZymes, InterPro IDs, Pfams). To these we assigned activity on the common plant cell wall substrates cellulose, hemicellulose, lignin and pectin following Miyauchi et al.[9].

**CAZymes**. Many CAZyme families are shared across all sampled genomes. Glycosyl transferases (GTs), which are involved in

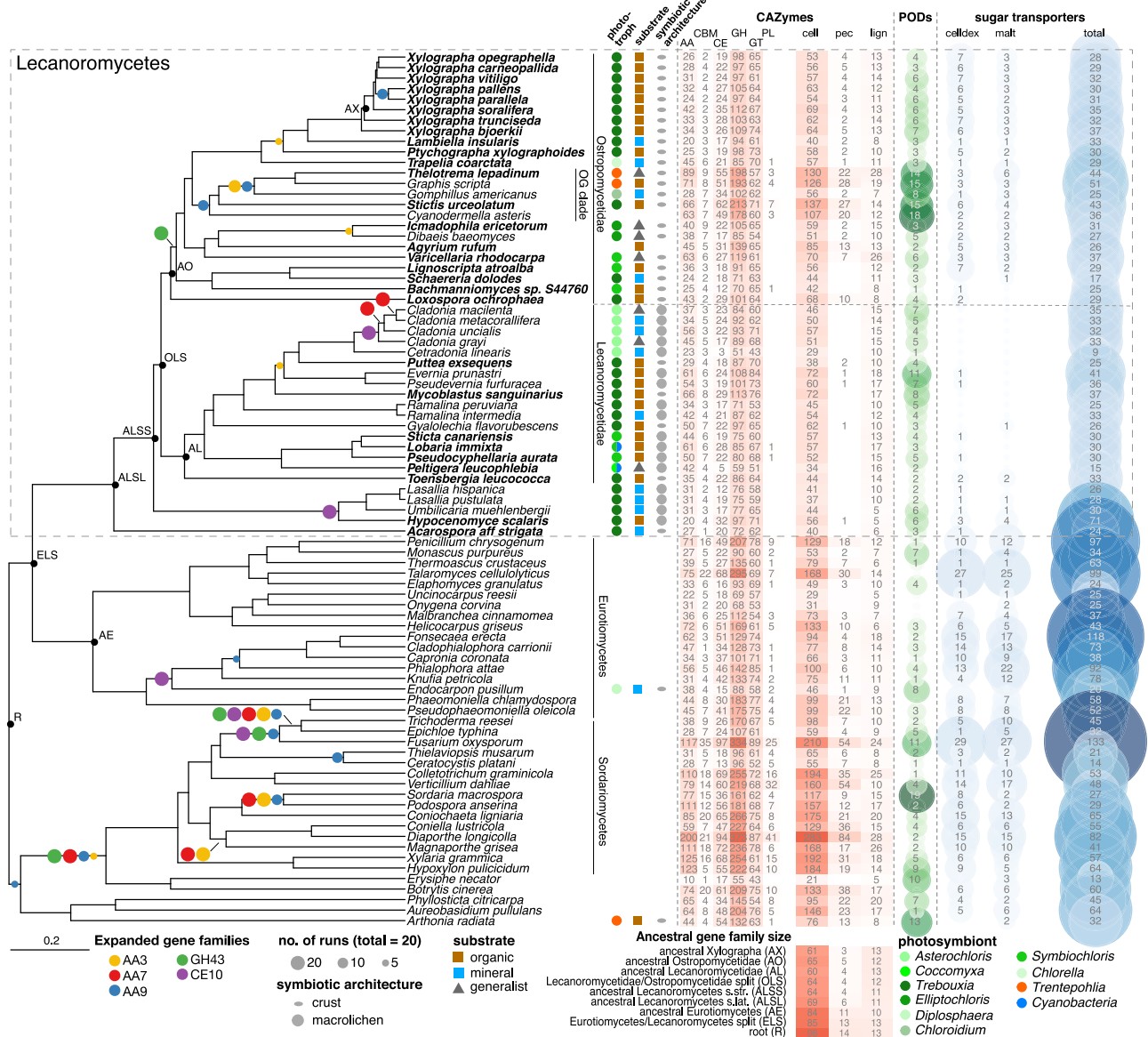

**Fig. 1 Distribution and ancestral states of CAZymes and selected sugar transporters across the evolution of *Lecanoromycetes* and related classes of *Ascomycota* projected onto a maximum likelihood phylogenomic tree based on 1310 loci.** Symbols beside tree tips refer to life history traits and phototrophic partners of LFS under study. Heatmaps with shades of red indicate the number of genes in different CAZyme classes or involved in degrading complex PCW components. Columns from left to right: AA Auxiliary Activities, CBM Carbohydrate binding module, CE Carbohydrate Esterases, GH Glycoside Hydrolases, GT Glycosyl Transferases, PL Polysaccharide Lyases. cell - Number of genes in 35 CAZyme families involved in cellulose and hemicellulose breakdown. pec - Number of genes in 11 CAZyme families involved in pectin breakdown. lign - Number of genes in three CAZyme families involved in lignin modification. Selection of CAZyme sets follows[9, 25]. PODs: numbers of heme haloperoxidase and DyP peroxidases potentially involved in lignin modification from Redoxibase (see text). Sugar transporters - selected PF00083 transporters. celldex - Number of cellodextrin transporters. malt - number of maltose transporters. total - Total number of other PF00083 transporters. Below the heatmap are ancestral sizes of CAZyme families involved in (hemi-)cellulose, pectin and lignin modification. Colored circles on tree branches indicate significantly expanded CAZyme families. The size of the circles indicates the number of individual CAFE runs (out of 20 total runs) in which a family was found to be significantly expanded. Expanded gene families recovered in less than five runs are omitted here. Exact numbers are given in Supplementary Fig. 12. Source data are provided as a Source Data file.

glycosylation and the synthesis of polysaccharides, differ little across all analyzed genomes and do not exhibit any significant reduction in Lecanoromycetes, suggesting that a core synthetic machinery remains largely unchanged across the evolution of the sampled Ascomycota (Fig. 1). While significant within-group variation exists, the mean number of glycoside hydrolase (GH) genes in lecanoromycete genomes is 40.7% lower compared to other sampled genomes ($p = 0.0000$; Supplementary Tables 9, 10 and 11). CAZymes with Auxiliary Activities (AA), Carbohydrate Binding Modules (CBM) and Carbohydrate Esterases (CE) are

also reduced significantly in Lecanoromycetes (Supplementary Table 10). The bulk of these differences can be attributed to a small number of CAZyme families (Supplementary Fig. 11), most of which are plant cell wall degrading enzymes (PCWDEs). Additionally three CAZyme families widespread in the sampled Ascomycota were not detected in Lecanoromycetes. These include PL4, which contains pectin degrading rhamnogalacturonan endolyases; CBM67, which binds to L-rhamnose and frequently occurs in multi-domain protein with enzymes in GH78 and PL1; and AA13, which contains lytic polysaccharide

monooxygenases (LPMOs) involved in starch breakdown. In contrast to these reductions, numerous CAZyme families are not reduced at all in Lecanoromycetes compared to other sampled Ascomycota. Indeed some, including those involved in degradation of endogenous fungal cell wall polysaccharides, such as GH128 and AA5, are even expanded in Lecanoromycetes (Supplementary Fig. 11).

All lecanoromycete genomes possessed genes encoding enzymes predicted to act on plant cell wall compounds, including cellulose and hemicellulose (e.g., GH5 and GH43) and lignin (AA1, AA2 and AA5). Depending on the symbiont configuration, these organic polymers may be produced by the phototroph partner and/or be exogenous to the lichen symbiosis (see Discussion). Principal components analyses (PCA) of CAZyme family numbers in the sampled genomes reveals differences in the amount of variation in CAZyme composition among lecanoromycete and other sampled ascomycete genomes (phylogenetically corrected PCA: Fig. 2; regular PCA: Supplementary Fig. 10). For CAZymes predicted to act on cellulose, hemicellulose and pectin, the number of predicted gene families varies less among lecanoromycete genomes than among comparable classes of Ascomycota (Fig. 2A, C), reflected in tight clustering in Lecanoromycetes versus wide scattering in other ascomycete classes. For lignin-modifying enzymes the variation is similar (Fig. 2B). Within Lecanoromycetes, however, the greatest amount of variation in predicted gene sets is exhibited in the subclass Ostropomycetidae (represented in our sample by 24 genomes), specifically in cellulose/hemicellulose (Fig. 2A) and pectin degradation (Fig. 2C), resulting in scattering in the PCA ordination. By contrast, genomes from the subclasses Acarosporomycetidae, Umbilicariomycetidae and Lecanoromycetidae form a cluster of similar predicted CAZyme sets, resulting in tight clusters in the PCA ordination. These latter three subclasses are represented in our dataset by one, four and 17 genomes, respectively; for convenience results reported for Lecanoromycetidae will refer by extension to results from all three of these subclasses unless otherwise specified.

Genomes from the Ostropomycetidae possess higher numbers of predicted CAZyme genes that act on cellulose and hemicellulose than those of Lecanoromycetidae, which are reflected in significantly higher numbers of GHs and CEs ($p = 0.0000$ and $p = 0.0024$ respectively; Supplementary Table 10). The differences are driven by one lineage in particular, represented by five genomes from the orders Ostropales and Gyalectales and referred to here as the OG clade (Fig. 1), and are much less if the two suborders are compared without the OG clade genomes. The OG clade contains numbers of predicted CAZyme genes for cellulose and hemicellulose modification that are over fourfold more numerous than the lowest lecanoromycete numbers, which are in *Cetradonia* and *Peltigera*, and equal or exceed the numbers in well-studied eurotiomycete saprotrophs such as *Penicillium*. These disparities are accounted for in large part by gene assignments to two CAZyme families, GH5 and GH43. Because both of these are large, heterogeneous families that include multifunctional CAZymes, we mapped putative GH5 and GH43 orthologs from the analyzed genomes against sequences of experimentally validated enzymes and predicted which are secreted (Fig. 3). Extensive gene duplication in Ostropomycetidae is found in gene sequences close to characterized cellulases (GH5 subfamily 5; EC 3.2.1.4) and from both lecanoromycete subclasses in sequences close to characterized 1,3-beta-glucosidases (GH5 subfamily 9; EC 3.2.1.58) and endo-1,4-beta-mannosidases (GH5 subfamily 7; EC 3.2.1.78). Notably, many putative lecanoromycete CAZyme genes from GH5 do not closely cluster with any characterized sequences and form their own clades with sequences from other sampled classes of Ascomycota (Fig. 3A).

The subclass Ostropomycetidae also possesses a larger proportional representation of genes coding for enzymes predicted to be involved in hemicellulose, specifically xylan, breakdown. Families GH6, GH7, GH11 and GH62, GH67 and GH131 were absent in Lecanoromycetidae and are only present in Ostropomycetidae in the OG clade and the saprotroph *Agyrium* (Supplementary Fig. 11). Hemicellulose breakdown also involves several CEs of which CE2 is present in the majority of Ostropomycetidae genomes and absent in Lecanoromycetidae (Supplementary Fig. 11). CE2 contains acetylxylan esterases (AXEs;[30]); acetylxylan is a major component of hemicellulose. An analysis of predicted GH43 gene sequences and their subcellular location compared to those of characterized enzymes shows that most characterized GH43s are in Ostropomycetidae (subfamilies 1, 6, 24, 26 and 36), but as with GH5s, the majority of sequences are not close to any characterized CAZymes (Fig. 3B).

Numerous enzymes can potentially be involved in lignin modification[17]. In Lecanoromycetes, CAZymes associated with lignin modification are predicted in all genomes, but again the highest numbers are found in Ostropomycetidae (Fig. 1). The numbers of predicted genes in *Graphis*, *Thelotrema* and *Varicellaria* are among the highest in the analyzed genomes. Most of these genes are members of CAZyme family AA1 (laccase, ferroxidase and multicopper oxidases, EC 1.10.3.x), and several are phylogenetically close to experimentally characterized enzymes from EC 1.10.3.2, which included laccase, p-diphenol:oxygen oxidoreductase and ferroxidase (Supplementary Fig. 8). We also found smaller numbers of predicted AA2 which contain Class II peroxidases (EC 1.11.1.x; Supplementary Fig. 9) in Lecanoromycetes, but none of these are phylogenetically close to any experimentally characterized peroxidases (Supplementary Fig. 15). As not all commonly studied enzymes involved in lignin modification are included in the CAZy database, we also mapped ortholog groups for all peroxidases in RedOxibase ([31], see Supplementary Fig. 6). Two ascomycete peroxidase groups that have been shown to have activity on lignin molecules, heme haloperoxidases and dye-decolouring peroxidases (DyP), exhibit large differences in predicted gene numbers in Lecanoromycetes (Fig. 1). Heme haloperoxidases occur in three ortholog groups, of which one was recovered in all sampled Lecanoromycetes but with greatly elevated numbers in the OG clade, as well as in *Pseudevernia* and *Mycoblastus* (Supplementary Fig. 6). A second heme haloperoxidase ortholog group is found in several Ostropomycetidae, while a third was not recovered in Lecanoromycetes. DyPs likewise occurred in three ortholog groups, one of which was recovered in most Ostropomycetidae and in about half of Lecanoromycetidae, and another of which was recovered only in the wood-dwelling genus *Xylographa* (Supplementary Fig. 6). The third DyP ortholog group was not detected in any lecanoromycete genome.

The largest differences in predicted PCWDEs among lecanoromycete genomes come from CAZymes acting on pectins. All but three of the Ostropomycetidae genomes possess a few predicted CAZymes involved in pectin degradation (Fig. 1). These include GHs (GH28, 49, 53, GH79, GH108), CEs (CE8, CE12, CE15) and the only polysaccharide lyases (PL; from PL1 and PL3, containing pectate lyases, EC 4.2.2.2) predicted in Lecanoromycetes (Supplementary Fig. 8). Lecanoromycetidae, by contrast, almost completely lack gene predictions for pectin degradative enzymes and completely lack PLs. Similar to CAZyme predictions for cellulose, hemicellulose degradation and lignin modification, the largest numbers of predicted genes involved in pectin degradation come from the OG clade of Ostropomycetidae. In these genomes, predicted gene numbers for CAZymes involved in pectin degradation even exceed those of known model saprotrophs in Eurotiomycetes, though the difference is not significant.

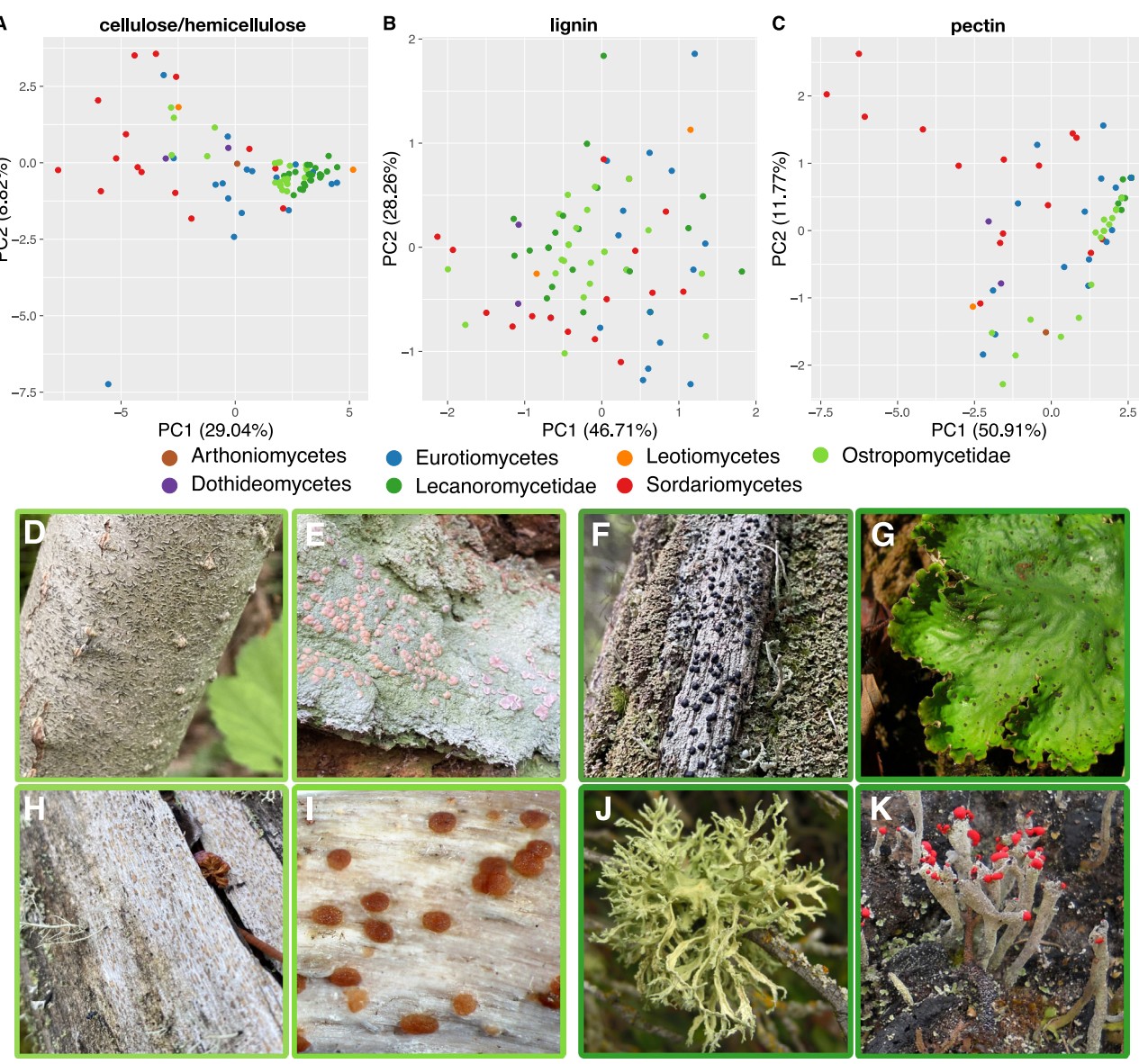

**Fig. 2 Similarity of CAZyme sets involved in the breakdown of different complex plant-based polysaccharides based on phylogenetically corrected Principal Components Analysis.** Different colors indicate taxonomic groups; genomes from the subclasses *Acarosporomycetidae* and *Umbilicariomycetidae* use the same color codes as *Lecanoromycetidae* for simplicity. **A** Similarity of CAZyme families involved in cellulose and hemicellulose breakdown. **B** Similarity of CAZyme families involved in lignin breakdown, **C** Similarity of CAZyme families involved in pectin breakdown. Displayed below are representative members of the two subclasses *Ostropomycetidae* (with light green border) and *Lecanoromycetidae* (dark green border). **D** *Graphis scripta*, **E** *Icmadophila ericetorum*, **F** *Mycoblastus sanguinarius*, **G** *Peltigera leucophlebia*, **H** *Xylographa carneopallida*, **I** *Agyrium rufum*, **J** *Evernia prunastri*, **K** *Cladonia macilenta*. Image credits: *Agyrium rufum*: Paul Cannon (fungi.myspecies.info); Creative Commons: BY-NC 4.0. *Peltigera leucophlebia*: Jason Hollinger, uploaded by Amada44, CC-BY 2.0, https://commons.wikimedia.org/w/index.php?curid=24213606. *Evernia prunastri*: by Jason Hollinger, CC-BY 2.0, https://commons.wikimedia.org/w/index.php?curid=50595319. *Cladonia macilenta*: Bruce McCune & Sunia Yang - Lichen, CC-BY 4.0-NC, https://lichens.twinferntech.net/pnw/species/Cladonia_macilenta.shtml; other images by the authors. Source data are provided as a Source Data file.

In addition to the PCWDEs outlined above, all sampled Lecanoromycetes possess a gene assigned to GH32, a family which includes invertases involved in the conversion of sucrose to glucose or fructose. Invertases are generally interpreted as an indicator of use of apoplastic plant sucrose. However, lecanoromycete GH32s do not cluster closely with any characterized invertases (Supplementary Fig. 13).

**Predictions of gene family contraction and expansion.** In order to establish whether CAZyme patterns in Lecanoromycetes are due to gene gain or loss, we reconstructed ancestral gene numbers

for each of the three main groups of PCWDEs (cellulose/hemicellulose, pectin, lignin). All three PCWDE groups are reduced already at the split between Eurotiomycetes and Lecanoromycetes (ELS node; Fig. 1), especially enzymes involved in (hemi-)cellulose and pectin breakdown. Much of the signal for this pattern comes from GH5 and GH43 which contain many well-characterized cellulases and hemicellulases, respectively (Fig. 3; Supplementary Fig. 7). Another apparent shedding of CAZymes, especially of those involved in pectin degradation, occurs concomitantly with the origin of Lecanoromycetes, the largest extant clade of LFSs (ALSS and ALSL nodes; Fig. 1). Here, too, the signal comes mainly from GHs. In several cases our ancestral state

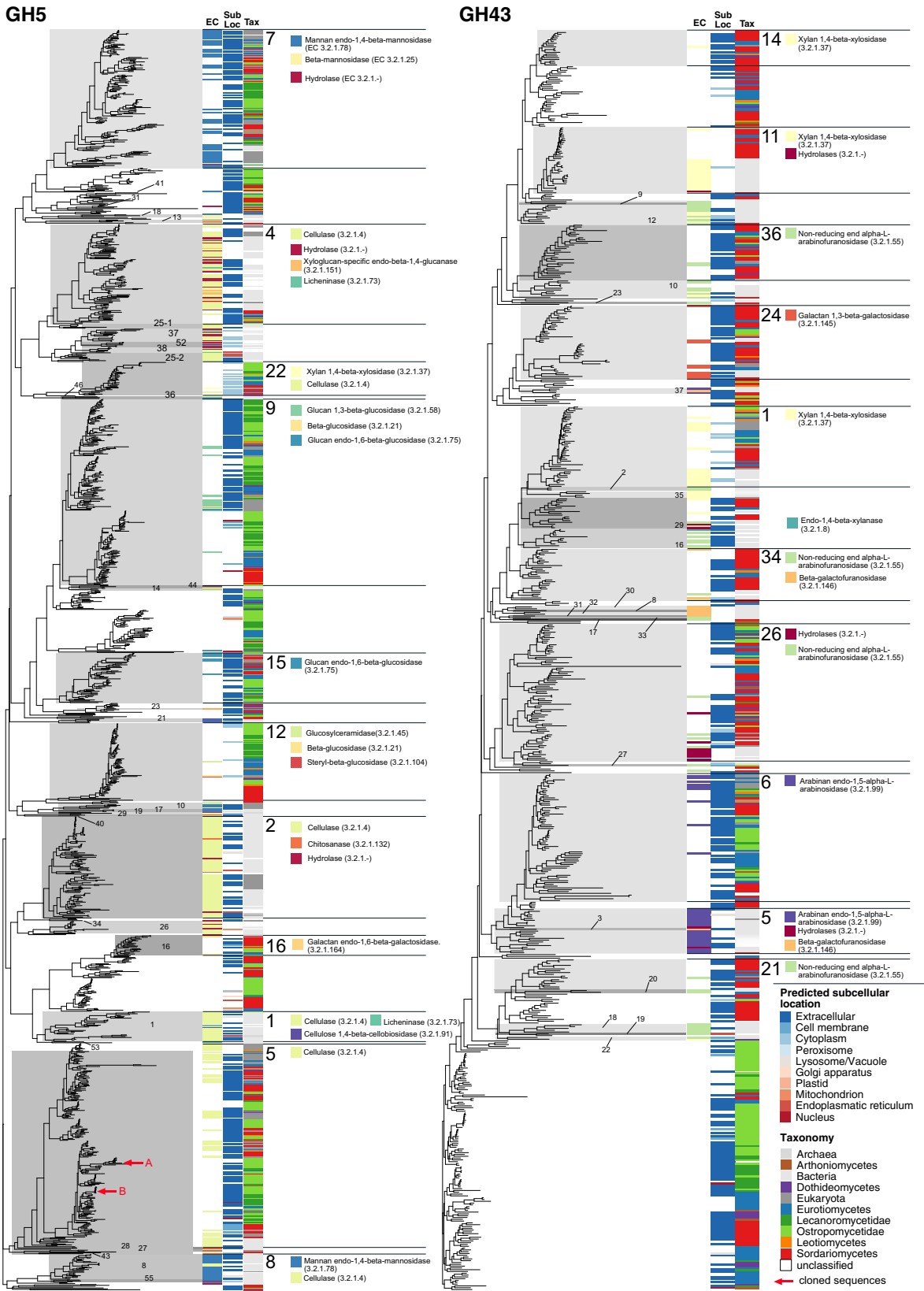

reconstruction indicates complete losses, despite the prediction of several of the relevant families in Ostropomycetidae genomes (Supplementary Fig. 8).

Gene family expansion analyses of all CAZymes revealed six significantly expanded families along different branches in the phylogeny. The only GHs expanded within Lecanoromycetes are GH43, which are expanded just after the last common ancestor of Ostropomycetidae (Fig. 1; Supplementary Table 8); another, GH18, was expanded in four model runs at the base of Sordariomycetes (Supplementary Fig. 12). Other significantly expanded CAZyme families in Lecanoromycetes include Auxiliary Activity (AA) families AA3, AA7 and AA9, which are

**Fig. 3 Gene trees of two CAZyme families involved in cellulose (GH5) and hemicellulose (GH43) breakdown. Each tree includes all experimentally characterized sequences combined with all sequences from the 83 genomes studied here.** CAZyme Subfamilies are labeled with numbers and gray rectangles. Three columns along tree tips display additional information of corresponding sequences when available. EC - Enzyme Code of experimentally characterized sequences downloaded from cazy.org. Sub Loc - Predicted subcellular location of Enzyme with DeepLoc. Tax - Taxonomic assignment of organisms from which the sequence comes from. For larger subfamilies, functions of characterized sequences based in Enzyme Code numbers are given as colored squares. Sequences used for heterologous expression experiments are marked with red arrows in GH5 subfamily 5. Source data are provided as a Source Data file.

variously expanded in *Xylographa*, the OG clade and *Cladonia*. Auxiliary Activity families typically act in concert with other CAZymes and the three expanded AA families all contain genes involved in cellulose breakdown. The only other expanded CAZyme family was CE10, now considered to be a family of esterases acting on non-carbohydrate substrates, which was expanded at the last common ancestor of *Cladonia* species and Umbilicariomycetidae.

**Transporters**. As an additional line of evidence for potential use of exogenous carbohydrates, we mapped the numbers of predicted sugar transporters across all genomes (Fig. 1). Most predicted transporters were more or less evenly represented across most genomes, but two groups of transporters exhibited distinct presence/absence patterns within Lecanoromycetes. Cellodextrin transporters, close orthologs of experimentally demonstrated cellodextrin transporters from *Aspergillus* and *Penicillium* (Supplementary Fig. 5), were predicted in all Ostropomycetidae genomes except *Schaereria*, and were most numerous in *Lignoscripta*, *Stictis*, *Ptychographa* and *Xylographa*, all of which are LFSs of lichens with high wood or bark specificity (Fig. 1). Cellodextrin transporters are involved in transmembrane import of cellobiose and other cellodextrins, which are short beta-linked glucose fragments of cellulose. Of the remaining 22 lecanoromycete genomes, all four Umbilicariomycetidae and the one Acarosporomycetidae genome had cellodextrin transporters, while only five of the 17 Lecanoromycetidae genomes did; in all of these, gene numbers were well below the average observed in Ostropomycetidae. We detected a similar pattern for the predicted occurrence of maltose transporters (Fig. 1), involved in transmembrane import of alpha-linked starch breakdown products: 22 of 24 Ostropomycetidae genomes had predicted maltose transporters, but only five lecanoromycete genomes outside of Ostropomycetidae had any, three of them in Umbilicariomycetidae.

**Evidence of cellulase functionality**. To validate the functionality of putative cellulases found in lichens, we selected *X. bjoerkii* genes *MMC18_000518* (MCJ1387675.1 and *MMC18_004565* (MCJ1391700.1) after aligning sequences from all *Xylographa* species recovered in GH5 Subfamily 5 (Fig. 4d), hereafter called *cellulase A* and *B*, for further analysis based on their similarity with Cel5A cellulase sequences from *Trichoderma reesei*, for which there has been substantial structural and functional characterization. The cellulase domain of each gene was expressed in *Escherichia coli* as a C-terminal fusion with the maltose binding protein, which enhances protein expression and stability and provides a convenient handle for purification. The partially purified proteins were active on both cellulose (β-1,4-linked glucose) and barley β-glucan (alternating β-1,4/β-1,3-linked glucose) but not xylan (β-1,4-linked xylose containing side branches of α-arabinofuranose and α-glucuronic acids) (Fig. 4a–c). Both exhibited a pH optimum of 5, similar to other Cel5A enzymes but differ slightly in their temperature optima. While both are active at 20 °C, only *cellulase A* is also active at higher temperatures. Closely related orthologs of *cellulase A* and *cellulase B* were

recovered in all lecanoromycete genomes. Lecanoromycetidae mostly only had a single ortholog with the exception of *Lobaria*, *Pseudocyphellaria*, *Pseudevernia* and *Sticta* which each have two orthologs (Fig. 4d). In contrast, the majority of Ostropomycetidae genomes have two or more orthologs, and the highest numbers can be found in *Xylographa* species and *Stictis* which each have five (Fig. 4d).

## Discussion

Our analysis of CAZyme and sugar transporter genes paints a picture of a lecanoromycete arsenal that is larger, more diverse and more consistently present than expected. Our data show a reduction in mean numbers of lecanoromycete degradative CAZyme genes relative to other sampled Ascomycota, but also reveal significant differences within Lecanoromycetes. Genes for the breakdown of cellulose, hemicellulose and pectin are disproportionately enriched in the subclass Ostropomycetidae, and some genomes, notably in the OG clade, possess overall CAZyme numbers and functionality similar to those of well-known model saprotrophs such as *Aspergillus* and *Penicillium*. Furthermore, LFSs associated with the same genus of algal phototroph, *Trebouxia*, can retain multiple genes coding for pectin degradation as well as cellodextrin transporters (in *Lambiella*, *Loxospora*, *Ptychographa* and *Xylographa*), or they can largely lack these genes (in *Ramalina*). This implies that association with *Trebouxia* does not in itself result in gene loss, and the exact nature of the fungal-phototroph interaction, and other aspects of symbiosis biology, may need to be reconsidered. Understanding when and where in the fungal mycelium the CAZymes are deployed will be critical to determining their biological significance.

The largest differences in CAZyme gene numbers among the sampled lecanoromycete genomes represent PCWDEs. This raises a central question: what are the targets of LFS PCWDEs? The two most obvious candidates are the phototroph itself, on the one hand, and lichen-exogenous polysaccharides, such as wood and tree bark, or even non-plant polymers, on the other. The first possibility — targeting of the alga — echoes suggestions in studies of ectomycorrhizal fungi, where PCWDEs have been postulated to play a role in "cell softening" or "remodeling" of root tissues in their vascular plant symbionts upon contact initiation[32]. Cellulases and other CAZymes have been postulated to be involved in haustorial penetration of the algal cell wall[33,34] and degradation of algal cell walls in fresh lichen growth tips[35]. They could also play a role in digestion of dead algal cell walls, especially if algal populations turn over during the lifespan of the thallus, as has long been suspected[36]. Transcriptomic studies of isolates, cocultures and natural lichens could provide evidence of this. In two studies that reported CAZyme differential expression in symbiont coculture experiments, one detected upregulation of multifunctional GHs that could also be involved in fungal cell wall modification (GH2 and GH12), but none of the core lecanoromycete cellulases or hemicellulases we report here[37], while another reported upregulation of gene models assigned to GH5, GH43 and GH45[38] (in their Additional Files 6.3 and 6.4). Most common algal symbionts are thought to contain cellulose in their

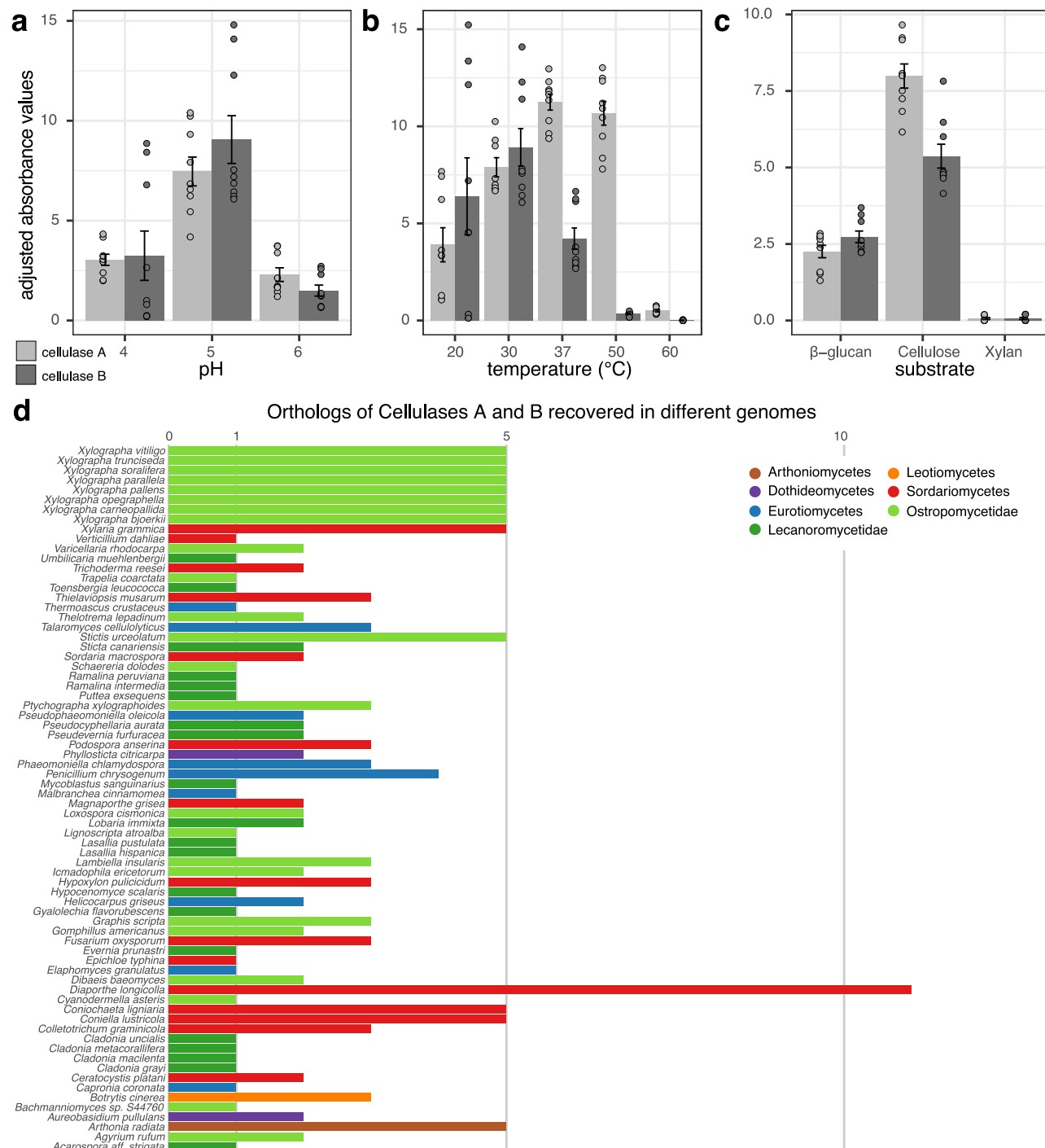

**Fig. 4 Enzymatic activity of two putative cellulases A and B from *Xylographa bjoerkii* and orthologs of these genes.** Activity of cellulases A and B at different pH conditions (**a**), at different temperatures (**b**), and with different substrates (**c**). Y axis represents mean absorbance ($+/-$ standard error of mean) at 595 nm after adjusting for the dilution factor. Experiments were performed in technical and biological triplicates. **d** Number of orthologs of cellulase A and B in the 72 genomes that possess them. Different colors indicate taxonomic groups; genomes from the subclasses *Acarosporomycetidae* and *Umbilicariomycetidae* use the same color codes as *Lecanoromycetidae* for simplicity. Source data are provided as a Source Data file.

cell walls but no pectin[39]; it is unclear if any lichen algal symbiont possesses pectin in its cell walls.

The second possibility, that LFS PCWDEs target lichen-exogenous polysaccharides, is supported by two lines of evidence. The first is experimental. Both cellulases and polygalacturonases, which are active on pectins, have been detected in incubated whole lichens both in the presence[40,41] and absence[42] of cellulose-containing algal photosymbionts. Cellulase production furthermore has been reported to vary depending on the species of tree it is growing on[43]. Some lichens have also been experimentally shown to possess an enzymatic and extracellular redox cycling toolbox consistent with lignin modification[21,44,45]. The second line of evidence relates to the ecological substratum use and relationship to known saprotrophs. The enrichment of PCWDEs we found tracks closely with the secondary origin of non-lichen saprotrophs and, among lichens, those with wood-

obligate ecology. Several secondary origins of putative fungal saprotrophs occur within the Lecanoromycetes, some of which include fungi involved in "optional lichens", in which the fungus can occur either as an LFS or saprotroph[46]. The most speciose of these saprotroph groups, the Ostropales, arose within our "OG clade" that possesses the largest PCWDE enrichment of all sampled lecanoromycete genomes. We also found cellulase gene family expansions in LFSs of wood-obligate lichens such as *Ptychographa* and *Xylographa*, and our experimental evidence for cellulase activity derives from one of these genomes (*Xylographa bjoerkii*).

Enzymatic targeting of algal cell walls or lichen-exogenous polymers are not mutually exclusive possibilities. Nor do all exogenous substrates necessarily have to be plant cell walls, as lichen symbionts also interact with a wide range of other fungi, notably fungal parasites and bacteria[47]. Determining where and when these genes are expressed in nature will require consideration of the life cycle and spatial extent of LFS mycelia, both of which are still poorly understood. The mycelia of sexually reproducing LFSs go through an aposymbiotic stage of unknown length, during which it has been suggested they may be saprotrophic[48]. Lichens at high latitudes or under snowpack may go through seasonal fluctuations in fixed carbon input, which could be augmented by other sources[21]. Even after symbiosis is established, parts of the mycelium can be free of phototroph cells, and exhibit deviations in carbohydrates that suggest other metabolic processes are in effect than in the phototroph-associated mycelium[49]. Many lichens include a phototroph-free "prothallus", in which fungal hyphae radiate beyond the zone of phototroph cells[50]. Others possess a "hypothallus", a phototroph-free cushion of mycelium in direct contact with the substrate[49]. Macrolichens are often anchored onto their substrates by phototroph-free "holdfasts", "rhizines" or mycelial pegs which have traditionally been interpreted as having an exclusively structural, stabilizing function, but can extend as mycelial networks into xylem[51] or living moss mats[42]. Parts of the mycelium with phototroph cells may also differ based on position relative to growth tips, which are thought to include larger proportions of living cells[52], and entire strata can be phototroph-free. We currently lack data on the role CAZymes could play in nutrient scavenging or cell wall remodeling in these phototroph-free sectors of the symbiont mycelium.

In addition to the differences and enrichment patterns we have analyzed here, the lecanoromycete genomes we surveyed are rich in genes that do not cluster with sequences from experimentally characterized CAZymes. Much work remains to be done to characterize their substrates and activity. These include two clades of GH5 cellulases (Fig. 3), numerous sequences assigned to GH43 hemicellulases but without closely clustering orthologs (Fig. 3), putative GH32 invertases (Supplementary Fig. 13), and numerous sequences assigned to AA5 (Supplementary Fig. 17), GH30 (Supplementary Fig. 38), GH72 (Supplementary Fig. 53), and GH79 (Supplementary Fig. 55). The need for more experimental work is especially true of putative lignin-modifying enzymes. Lichen fungi have been shown to possess elements of a lignin modification toolbox including a brown rot-like system of hydroxyl radical production in the presence of a quinone and chelated Fe[44,53], as well as a variety of metal-containing oxidoreductases including laccase and heme peroxidases that are induced by starvation and treatment with soluble cellulose and lignin breakdown products[21,44]. Specifically, experimental work has shown that *Lsa*POX, a putative heme peroxidase from the fungal symbiont of the lichen *Leptogium saturninum*, can metabolize the β-O-4 lignin model dimer adlerol into veratraldehyde, a reaction also performed by white-rot fungi[45] (a published oligopeptide sequence of *Lsa*POX is closest to 4206_AniHalPrx05 in

Redoxibase, with a close hit in *Cladonia grayi*, corresponding to the haloperoxidase_.haem. column in Supplementary Fig. 6). Our survey shows that laccase and peroxidase ortholog groups are widespread in Lecanoromycetes and especially enriched in species on wood, with numerous novel sequence groups. However, none of these alone should be interpreted as *prima facie* evidence of lignin modification in nature, as these enzymes can also be involved in other aspects of lichen biology, such as degradation of secondary metabolites[44].

Did stable phototroph association coincide with the loss of PCWDEs in LFSs? Looking only at the reductions in mean CAZyme gene numbers, the answer would appear to be positive, but the occurrence of LFSs in the OG clade (four of which in our sample are LFSs and one of which, *Cyanodermella*, is an endophyte) — with CAZyme arsenals as large or larger than those of many saprotrophs — shows that LFSs do not necessarily lack CAZyme arsenals. Despite their minority representation in our data set, CAZyme-rich LFSs may in fact be numerous in nature: the Ostropales and Gyalectales which make up the OG clade include over 3200 named LFS species, almost 17% of named LFSs[54], in addition to approximately 500 named non-LFS fungi, mostly saprotrophs. For phototroph acquisition to have no evolutionary consequences for the CAZyme arsenal, symbiont-derived photosynthates would likely have to provide a function other than as a substrate for growth and respiration. In fact this has already long been postulated, in the form of a role for polyols as compatible solutes[55]. Some have even suggested this may be the main role for transferred photosynthates[56,57]. A further indication that photosynthates are not solely nutritional subsidies could be the retention of invertases in Lecanoromycetes. In ectomycorrhizal fungi, the loss of invertases is thought to limit their ability to access plant sucrose and reinforce their dependence on plant-derived glucose[9]. Though the orthologs we found did not exhibit high similarity to characterized sequences, the prediction of an invertase is consistent with the ability to culture LFSs on sucrose[58] and experimental evidence of invertases in lichen fungi[20,59].

A strong inference about whether LFS CAZyme gene reduction coincided with the onset of stable phototroph association requires greater certainty about the ancestral states along the ascomycete backbone, and the lecanoromycete phylogenetic backbone in particular. Our ancestral state reconstruction suggests a gradual loss of GHs and PLs since the last common ancestor of Lecanoromycetes and Eurotiomycetes, and only pectin degradation genes are inferred to have been abruptly reduced. This reconstruction is sensitive to taxon undersampling and may be a conservative estimate. The large CAZyme arsenal of the OG clade, with dozens of unlinked genes and numerous PLs, is unlikely to have been acquired by lateral gene transfer. If the PL-rich OG clade CAZyme arsenal is in fact ancestral, this would imply that CAZyme loss is not an automatic consequence of stable association with phototroph symbionts, but rather of subsequent events or adaptations. If so, this implies that the OG clade CAZyme arsenal was lost no fewer than seven times in our tree. Support for a scenario of frequent mass gene reduction comes from the fact that one such loss appears to have happened within the OG clade itself: *Gomphillus* exhibits marked gene number reductions compared to its closest sampled relatives. Our data do not currently allow the hypothesis that the OG clade CAZyme arsenal is ancestral to be rejected. If it is not, it becomes more likely that CAZyme loss in LFSs is driven by additional processes.

Lecanoromycetes, the focus of our study, are the largest extant group of LFSs. A fuller picture of CAZyme arsenal retention during lichen symbioses and especially in pairing with specific phototroph partners will emerge as more genomic data become

available from other fungal classes. Two additional LFS genomes were included in the outgroups of our dataset, *Endocarpon* (Eurotiomycetes) and *Arthonia* (Arthoniomycetes). Both genomes echo the lack of predicted cellobiose transporters found within numerous Lecanoromycetes, but differ in their predicted pectin degradation capacity, with *Arthonia* possessing gene numbers similar to those found in the Ostropomycetidae. *Arthonia* also possesses higher numbers of non-CAZyme peroxidases, similar to the OG clade. If the OG clade CAZyme arsenals are related to saprotrophy, similar CAZyme arsenals could be expected in lichen fungal symbionts e.g., from the Dothideomycetes, which have long been postulated to have dual lifestyles as mutualists and saprotrophs[60]. Within Lecanoromycetes, the most striking apparent functional losses, both in terms of CAZyme genes as well as in cellodextrin and cellobiose transporters, occurred outside of the Ostropomycetidae, especially in the subclass Lecanoromycetidae, which largely lacks these two types of transporters. LFSs in Lecanoromycetidae, like those in Ostropomycetidae, are considered obligate symbionts of algae or cyanobacteria. They share many of the same algal symbionts, especially *Trebouxia*, and have no known lifecycle differences. However, they differ in general thallus architecture. Ostropomycetidae almost exclusively form crustose thalli in which thallus-to-substrate surface area contact is maximized. Lecanoromycetidae, and to some extent Acarosporomycetidae and Umbilicariomycetidae, include LFSs involved both in crustose thalli as well as so-called macrolichens, in which the thallus often becomes greatly expanded into leaf-, hair- or shrub-like forms, in which surface contact is minimized. Lecanoromycetidae macrolichens include some of the largest lichens by biomass, and in theory these would require more carbon, not less. So what additional adaptational processes could drive CAZyme loss?

Two basic biological traits surrounding the crust-to-macrolichen transition deserve more attention with respect to their potential effect on sugar uptake. In the first, the type of fungal-algal contact differs, from intracellular haustoria in many crust lichens to so-called intraparietal haustoria, which do not breach the cell wall, in some crusts and all macrolichens[61]. The exact consequence of this is unknown, but intracellular haustoria are characteristic of pathogenic fungi and may require a greater variety of CAZymes to penetrate the algal cell wall. In the second, macrolichens owe their architecture to a well-developed exopolysaccharide gel, termed cortex, which forms a rigid structural scaffold considered a prerequisite to macrolichen formation[62]. The polysaccharide composition of this layer varies widely across lichen symbioses[63] and is poorly characterized and mapped across the phylogenetic tree. Additional evolution in cortex layering happened in many lichens involving Lecanoromycetidae[64], though in how many remains unclear. The cortex, which is hydrophilic, also mediates water retention[63] and has been shown to operate like a sponge for passive uptake of dissolved nutrients[65] and glucose[66]. One of the largest epiphytic macrolichens has been experimentally demonstrated to take up tree-derived glucose[67]. Whether this is specifically facilitated by the cortex remains to be tested, but as the cortex mediates the passage of environmental molecules for many macrolichens, it seems likely. If capture of simple sugars is found to be a general function of the cortex in different environments, it could be expected to have significant evolutionary consequences for maintenance of CAZyme genes used for more costly carbohydrates.

The existence of a robust carbohydrate breakdown machinery across a large swathe of LFS evolution calls into question the assumption, underlying decades of ecophysiological work on lichens, that the lichen carbon economy is exclusively the sum of algal $CO_2$ fixation. The finding that some LFSs have CAZyme arsenals on par with saprotrophs lends support to Schwendener's hypothesis that different types of lichens exist: those with fungal symbionts that depend mostly on their phototroph, and those with fungal symbionts that augment their carbon assimilation from external sources[1]. The longstanding assumption that LFSs solely utilize fixed carbon must now be weighed against competing hypotheses, including A) that some or many LFSs build their mycelia from non-algal carbon sources, including absorbed monosaccharides and complex polysaccharides, before or during symbiosis; and B) that many different models of carbon acquisition—fixed, seasonal, facultative, scavenged and/or absorbed—may exist under the umbrella of what we currently call lichen symbiosis.

## Methods

Extended Materials and Methods as well as Supplementary Figures and Tables are available as Supplementary Information. All used software is listed in Supplementary Table 3.

**Used genomes**. We built a data set consisting of 83 fungal genomes from the phylum Ascomycota, including 46 genomes from the Lecanoromycetes, twenty-nine of which were newly generated for this study, and 37 genomes from the related classes Eurotiomycetes, Dothideomycetes, Arthoniomycetes and Sordariomycetes. Within Lecanoromycetes, the acquisition of new genomes was targeted to include a representation of lineages including 1) LFSs of macrolichens (*Cladonia*, *Evernia*, *Lobaria*, *Peltigera*, *Pseudevernia*, *Pseudocyphellaria*, *Ramalina*, *Sticta*, *Umbilicaria*); 2) LFS lineages involved in crust-forming lichens for which carbon acquisition from the substrate has been postulated, including wood specialists (*Lecidea scabridula*, *Lignoscripta*, *Ptychographa*, *Puttea*, *Xylographa*[18]), bark specialists (*Graphis*, *Loxospora*, *Schaereria*, *Stictis*, *Thelotrema*, *Varicellaria*), specialists of decaying plant matter (*Gomphillus*, *Icmadophila*), mineral soil (*Dibaeis*), rock (*Acarospora*, *Trapelia*) and a lichen-on-lichen "parasite" (*Lambiella*); 3) LFS lineages from crust-forming lichens that behave as ecological generalists (*Gyalolechia*, *Mycoblastus*, *Toensbergia*); and 4) a lecanoromycete saprotroph (*Agyrium*) and endophyte (*Cyanodermella*[68]). Our sampling simultaneously represents a cross-section of lecanoromycete evolution, with 24 genomes from the subclass Ostropomycetidae, 17 from the subclass Lecanoromycetidae, and one each from the species-poor subclasses Acarosporomycetidae and Umbilicariomycetidae, respectively. Ten genomes were derived from cultured samples and 18 are metagenome-assembled genomes (MAGs) assembled and binned according to Tagirdzhanova et al.[69]. Culture-derived genomes differed little from MAGs in estimated completeness and gene numbers (Supplementary Figs. 1 and 2).

**Genome assembly and filtering**. Raw sequence data was inspected with FastQC 0.11.7 and trimmed with trimmomatic 0.29 to remove adapter remnants and low-quality reads. Trimming parameters are given in Supplementary Table 4. Genomes were assembled using SPAdes 3.12.0 or Abyss 2.0.2. Assembly parameters are provided in Supplementary Table 5. To extract LFS contigs, draft assemblies were filtered using blobtools 1.1.1 or CONCOCT 1.2 and assembly completeness assessed with Quast 4.6.3 and BUSCO 3.0.2. We additionally extracted mitochondrial contigs by blasting mitochondrial genes downloaded from NCBI against each de novo assembly. We then discarded contigs with blast hits of e < 1e-03 and alignment length >500 bp.

**Gene calling and functional annotation**. We used funannotate 1.8.3 to perform gene-calling and functional annotation for all used genomes in the same way. This reduces potential biases introduced by different gene-calling and annotation methods. De novo sequenced genomes were repeat-masked using RepeatModeler and RepeatMasker. We then used Augustus 3.3, snap 2013_11_29, GeneMark-ES 4.62 and GlimmerHMM 3.0.4 and tRNA-Scan 2.0.5. Functional annotations for all predicted protein sequences were inferred using Interproscan-5.48–83.0, HMMer3 searches against dbCAN (v9) and Pfam (v33.1) as well as eggnog-scanner searches against EggNOG (v4.5.1) databases.

**Phylogenomics**. Phylogenomic trees were calculated using the phylociraptor pipeline (https://github.com/reslp/phylociraptor). We ran BUSCO 3.0.2 on each genome assembly to identify single copy-orthologous, combined amino-acid sequences of each BUSCO gene from all genomes. Only genes which were present in >50% of genomes were aligned using mafft 7.464 and trimmed using trimal 1.4.1. We calculated single-gene trees using iqtree 2.0.7 and a species tree using ASTRAL 5.7.1. We created a concatenated alignment from all alignments, estimated the best substitution model for each and calculated a tree based on a partitioned analysis of the concatenated alignment using IQ-Tree 2.0.7. We used the concatenated phylogeny to generate an ultrametric tree using r8s 1.81. This tree was used for subsequent analyses. We used custom python and R scripts to plot phylogenomic trees.

**Selection of CAZyme groups**. We selected sets of CAZyme families involved in (hemi-)cellulose, pectin and lignin degradation based on previous studies[9,25] (Supplementary Table 6). For the lignin set we additionally identified class II peroxidases based on similarity of all Ascomycota class II peroxidases downloaded from RedOxiBase (30; accessed Jul. 14, 2021; http://peroxibase.toulouse.inra.fr/) using Orthofinder 2.5.2.

**Ancestral state reconstruction of CAZyme families and similarity of CAZyme sets**. We reconstructed the ancestral size of each CAZyme family using CAZyme counts from our genome annotations and our ultrametric phylogenomic tree in R. We used anc.ML from phytools (v0.7–70) under an Ornstein-Uhlenbeck model of trait evolution. We then visualize ancestral states with custom R scripts.

We calculated phylogenetically corrected PCAs in R as implemented in the phytools function phyl.pca using maximum-likelihood optimization and regular PCAs using the prcomp function (Supplementary Fig. 10). We used log-transformed gene-counts of CAZyme families and our ultrametric phylogeny as input for all PCAs. The PCAs were visualized using custom R scripts.

**Gene family expansion analysis**. We analyzed gene family expansions of CAZyme families, based on CAZyme family count data from funannotate annotations and our ultrametric phylogenomic tree using CAFE 5 (git commit 08d27a1). We ran CAFE twenty times while accounting for different gene-birth rates and error-models and summarized the results using custom R and python scripts.

**Additional characterization of CAZymes**. CAZyme families known to contain important degradative enzymes (e.g., cellulases, hemicellulases) were further characterized using Saccharis v1 (git commit 9a748be; Supplementary Figs. 14–61). For each CAZyme family, we downloaded all characterized genes from cazy.org (accessed: March 3, 2021) including additional information such as taxonomy, accession numbers and CAZyme (sub-)family assignments. Using Saccharis, we created MUSCLE 3.8.31 alignments for each gene family of all (from genomes used in this study plus characterized genes from cazy.org) genes and created maximum-likelihood trees using Fasttree 2.1.10. For all genes included in our phylogenetic reconstructions, we also predicted subcellular locations using DeepLoc 1.0. We only considered predictions for subcellular locations if the predicted probability was >70%. We used custom R and python scripts to visualize the trees.

**Heterologous expression of putative cellulases**. Sequences assigned to GH5 subfamily 5, with confirmed cellulolytic activity, from obligately wood inhabiting LFS *Xylographa* species were aligned to the characterized and crystalized cellulase domain of *Trichoderma reesei* (PDB: 3QR3; https://www.rcsb.org/structure/3QR3). Based on sequence similarity, we selected two candidate cellulase sequences from *Xylographa bjoerkii* for testing cellulolytic activity.

Enzyme activity of the two candidate cellulases A and B was tested by combining in a 1.5 mL microcentrifuge tube 50 μL enzyme; 100 μL buffer A at either pH 4, 5, or 6; and 50 μL either AZCL-HE-cellulose, AZCL-β-glucan, or AZCL-xylan. Tubes were incubated at 4, 20, 37, 50, or 60 °C for 48 h. To measure activity, samples were centrifuged at 13,000 rpm for 5 min to settle any debris, then 100 μL supernatant was removed to a 96-well flat-bottom plate. Absorbance at 595 nm was measured by plate reader and blanked with a sample containing water instead of enzyme.

**Identification of sugar- and sugar-alcohol transporters**. To identify putative sugar- and sugar-alcohol transporters we used Orthofinder 2.5.2 on all sequences from all genomes with Pfam annotation PF00083 (Sugar_tr; http://pfam.xfam.org/family/sugar_tr; accessed August 5, 2021) combined with characterized sugar transporter sequences from the PF00083 seed set. Additionally, we added characterized cellodextrin (MH648002.1 (NCBI; from *Aspergillus niger*), S8AIR7 (UniProtKB; from *Penicillium oxalicum*)) and sugar alcohol transporters (AAX98668.1; from *Ambrosiozyma monospora*, CAR65543.1, CAG86001.1; from *Debaryomyces hansenii*, NP_010036.1; from *Saccharomyces cerevisiae*). We used the presence of characterized sequences in the inferred orthogroups to identify orthologs in each genome included in this study. The number of orthologs of different transporters per genome were visualized using custom R scripts.

**Identification of class II peroxidases**. First, we used diamond 0.9.22 to search all Ascomycota class II peroxidases downloaded from RedOxiBase (accessed Jul. 14, 2021) against the predicted proteins in all 83 genomes studied here. We then extracted all predicted protein sequences which had a diamond hit to any of the downloaded sequences. Similar to the identification of sugar transporters we used Orthofinder 2.5.2 to classify putative class II peroxidases present in our genomes based on the presence of downloaded genes in individual orthogroups. The number of orthologs per genome for different peroxidases were visualized using custom R scripts.

**Reporting summary**. Further information on research design is available in the Nature Research Reporting Summary linked to this article.

## Data availability

De novo generated genome assemblies and corresponding functional annotations are deposited at NCBI under BioProject PRJNA795879. Accession numbers for all genomes used in this study are provided in Supplementary Data 1. Used PFAM and RedoxiBase sequences, annotations of previously sequenced genomes, alignments and phylogenomic trees are available for download in a public Dryad repository (https://doi.org/10.5061/dryad.3xsj3txjb). Source Data are provided with this paper.

## Code availability

The analysis workflow used to acquire the results in this paper, including all custom python and R scripts, is available on Github[70] (https://github.com/reslp/LFS-cazy-comparative; https://doi.org/10.5281/zenodo.6453144).

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

## Acknowledgements

The early phases of this project were funded by the Austrian Science Fund (FWF grant P25237, "Evolution of Substrate Specificity in Lichens") and carried out at the Institute of Plant Sciences (Uni Graz). PR would like to acknowledge the Theodor Körner Funds (Vienna, Austria) and Network of Biological Systematics Austria (NOBIS; Vienna, Austria) for funding that enabled genome sequencing of several species. PR would also like to thank Christoph Hahn and Fernando Fernández-Mendoza for many methodological discussions. SW received funding from the Icelandic Research Fund IRF (174307-051), from Deutsche Forschungsgemeinschaft DFG (WE 6443/1-1) and from LMU Munich (startup funds). MW received funding from the Swedish Research Council, grant VR 2016-03589. MW and MK would like to acknowledge support from the NRM Department of Bioinformatics and Genetics, the National Genomics Infrastructure in Stockholm, the SNIC/Uppsala Multidisciplinary Center for Advanced Computational Science and the UPPMAX computational infrastructure, and Linda Phillips for assistance with material. TS would like to acknowledge an NSERC Discovery Grant, a Canada Research Chair in Symbiosis, and the generosity of Susan Dalby and Eskild Petersen, who provided a place to work on this manuscript. Special thanks go to Sophie Dang at the Molecular Biology Service Unit, University of Alberta Department of Biological Sciences, for help in data acquisition, and members of the U of A Lichen Evolution Lab for reading and commenting on the manuscript. We are also grateful to Sigrun Kraker, Theodora Kopun, Tanja Ernst and Andrea Brandl for laboratory assistance.

## Author contributions

T.S., P.R. and L.M.W. designed this study. Genomes used for this study were generated in the labs of T.S., H.M., J.P.M., S.W. and M.W. G.T. performed metagenome filtering;

S.F.R. assembled genomes of *Sticta*, *Pseudocyphellaria* and *Lobaria*, P.M. for *Xylographa* species; S.F.B., M.K. and L.M. organized cultures and genome sequencing for *Stictis*; S.W., H.G. and O.S.A. contributed the *Peltigera* genome; P.R. assembled all other genomes. A.R.B. performed heterologous expression of *Xylographa* cellulases. P.R. performed gene-calling, functional annotation and comparative genomic analyses. T.S. and P.R. wrote the manuscript with contributions from all authors.

## Competing interests
The authors declare no competing interests.
