## [Peer Review File · Nature Communications]

Reviewers' Comments:

Reviewer #1:

Remarks to the Author:

The manuscript of Resl et al. deals with the carbohydrate degradation and transport potential in the genomes of lichen fungal symbionts. The gene inventories for plant cell wall degrading enzymes (PCWDE) across 83 ascomycete genomes including lichen fungal symbionts (LFS) and non-symbionts (29 of the genomes newly sequenced) are compared. The authors start from the hypothesis that LFS in stable associations with their phototroph symbiont would show functional losses in PCWDE, similar to what has been found for ectomycorrhizal fungi. From their comparison they find some loss in enzymes acting on pectin, but overall all LFS retain a robust set of PCWDEs and some even have comparable gene copy numbers as well-known saprotrophic fungi, suggesting in most cases the remaining capacity to degrade cellulose and hemicellulose and to be not only dependent on the phototroph partner for carbon supply. Two predicted GH5_5 from wood inhabiting *Xylographa* species were further characterized for their cellolytic activity using heterologous expression. To test the potential use of exogenous carbohydrates by LSF via sugar transporters, the authors also compared the gene content for these transporters across all genomes and highlight the absence of Cellodextrin and maltose transporter in *Lecanoromycetes*.

Overall the data are well presented, the manuscript is very clear written and contains all necessary information and it's a very interesting topic.

Here my comments/suggestions:

My main concern comes from the lack of gene expression/transcriptomic/metatranscriptomic data (or other functional omics data like proteomics, but probably more difficult to generate). I don't understand why the authors haven't coupled the genome comparisons to "in lichen" gene expression data? The potential (presence of one or more copies of a gene) is one information. Gene regulation in the presence of the phototroph partner would be another possibility. In particular for "optimal lichen", a down-regulation of PCWDE as shown for example for the dark septate *Acephala* while forming ectomycorrhiza with pine (Miyachi et al. 2020) could be a possibility. Or the expression of only a subset necessary for symbiosis establishment, as it has been shown for most ectomycorrhizal fungi? The authors discuss most of these possible strategies, but there remains a lot of speculation and transcriptome data might provide first clues. This information would add additional value to the findings, even if it remains descriptive.

I further wonder why the authors focus on PCWDEs only? Carbohydrates could come from microbial cell walls (bacteria, fungi)? Any hints for this?

P5 110 mycorrhizal fungi- true for arbuscular and ectomycorrhiza, but not for ericoid or orchid mycorrhiza

P6 125 How did the authors select for the genomes to compare with? Please explain, give your criteria? This selection will somehow influence the outcome of your analysis?

P9 207 This expansion of GH5_5 and GH43 in *Ostropomycetidae* is very interesting. Again, it would be interesting to see if they are all expressed/functional genes, in particular the sequences not close to characterized Cazymes.

P10 217 So is there really any lignin-degradation potential?

P13 284 Please add here the Cazyme family (GH5_5) to make it clearer. How the two candidate genes were selected? Why two GH5_5 and not also one of the expanded GH43?

Supplemental Table 1 Please make sure that you've the authorization for non-published genomes. Perhaps also adding the respective publications, if published? Does the ??? for the new genome accessions mean that submission is in progress?

Reviewer #2:

Remarks to the Author:

The manuscript focused on the ability of lichen fungal symbionts to degrade and transport carbohydrates based on the genome sequences. In addition to the already available genome sequences, 29 new genomes were produced to evaluate the evolution of PCWDE encoding genes. The work shows that in lecanoromycete species there are more genes encoding CAZymes and sugar transporters than previously expected.

This work builds on the previous papers of the authors, being the largest data set studied so far. Although the mechanism of carbon acquisition in lichen symbiosis is still not fully explained, the work suggests diverse models based on the comparative CAZyme arsenals in LFSs.

The work is mainly based on the comparative genomic data. I wonder why the functionality of cellulases was validated by expressing two candidate enzymes only from GH5 and none from GH43, which was also expanded CAZy family. Also, examples of recombinant cellulases from species having different life style groups would have given better insight of the functionality of LFS cellulases. To obtain better understanding how functional the expanded gene families are, wider diversity of enzymes is needed. Otherwise, the conclusions remain speculative.

The definition of lignin degrading enzymes based on the AA families should be better explained. Unfortunately, CAZy database does not provide subfamilies in AA2. Ascomycetes do not have lignin degrading peroxidases i.e LiP, MnP and VP, however genes encoding DyPs, which have been shown to degrade mono- and dimeric lignin model compounds, as well as generic peroxidases are present in ascomycete genomes. The role of laccases in lignin degradation is also controversial. I suggest to use "lignin modifying enzymes", "lignin modification" and ability to degrade "lignin-derived compounds" to avoid misunderstanding.

The methodology is sound with the adequate level of details and it meets the expected standards in the field. However, Figure 4 needs modifications. Figure legend should be rephrased, because Fig. 4 does not show enzyme activities, but absorbance values at 595. Also include, how many replicates have been measured to get the standard deviation. I strongly recommend to convert absorbance values to enzyme activities either in units or katal. Y-axis shows Abs. 595 nm with values >4, which must be over the upper limit of the spectrophotometer reducing the reliability of the results. X-axis needs revision: No need to repeat pH or °C after every value, and there is a typo in pH. Please, use a symbol in β -glucan.

In the supplementary data: Is the Figure legend 11 correct? From Figure 13 on, the text above coloured bars is not readable or only part of the text can be seen. The taxonomy bar, where different taxonomical levels are mixed, is confusing. F. ex. Most lignin modifying peroxidases and laccases (AA1_1) are grouped under Eukaryota, where also all the other presented taxonomical levels belong to. Why not to use Basidiomycota? Please, check also the other figure legends. F.ex. Fig. 27, should only include genes for hemicellulose breakdown for GH10, Fig. 15 for lignin breakdown, CBMs should not include genes for lignin modification etc.

Minor points:

Line 161 and lignin (AA1, 2 and 5).

Line 162: Please, rephrase since lignin is not a carbohydrate.

Lines 195 and 207: In the Figure 3 legend, Figures 3A and 3B are not mentioned.

Line 244-245: Please, check if referring to the Figure 2 is correct. There is no indication to GH5 and GH43 in Figure 2.

Supplementary Tables 9-11: Please, rephrase the titles beginning with "This table shows..." with brief and specific description.

Reviewer #3:

Remarks to the Author:

This is a very interesting comparative genomic study designed to determine if the number of genes coding for plant cell wall-degrading enzymes (PCWDE) are lower in lichen-forming fungi, due to their specialization on green algae or other photobionts as the potential main source of carbon. The focus of this study is on Lecanoromycetes, which is the class of fungi with the largest number

of lichen-forming species. They compared 83 genomes distributed among six classes of fungi within the Leotiomyces (Pezizomycotina, Ascomycota), with most of the genomes representing the Lecanoromycetes (46 species), Eurotiomycetes (17 species) and Sordariomycetes (15 species). Among these 83 genomes, 29 (all belonging to Lecanoromycetes) have been newly sequenced for this study. The main result is that there is a reduction in number of some gene categories coding for CAZymes, PODS and sugar transporters, however this was far from being uniform among Lecanoromycetes, many of which have high numbers of PCWDE coding-genes in the range of fungal saprotrophs. This high variation of PCWDE encoding genes among Lecanoromycetes seems to be associated with the substrate on which these fungi grow (bark, wood, soils, bryophytes, or rocks), the thallus architecture (i.e., level of contact [or attachment] of the thallus with the substrate) and if the fungus loss or partially loss the ability to form a lichen symbiotic interaction (*Stictis urceolatum*, *cyanodermella asteris*, and *Agyrium rufum*). The main result is that some lichens may be obtaining carbon not only from their photobionts but also from the breakdown of lichen-external carbohydrates.

I have two main concerns about this study: (1) the taxon sampling and the overstatement of some of the results and (2) the emphasis of the discussion and conclusion on specific aspects of the study but not others to better fit the narrative of the manuscript.

(1) Throughout the study, the authors report their results as being across Ascomycota. Unfortunately, this is far from being the case. The sampling of this study focuses on one of three subphyla of Ascomycota, i.e., Pezizomycotina. Within Pezizomycotina, they sampled six classes (mostly three classes), which are all found within the Leotiomyces (Schoch et al. 2009, Systematic Biology). Therefore, the result of this study only applies to the Leotiomyces, at the very most, but more specifically to the Lecanoromycetes. Even within the Leotiomyces, no species from the Leotiomyces (5587 species, Kirk et al. 2008), Xylonomycetes (few species), Laboulbeniomycetes (2072 species), and Lichinomycetes (350 species all lichen-forming and closely related to the Lecanoromycetes and Eurotiomycetes) were included. Yet, in the genome-scale phylogenetic study of all fungi (Li et al. April 2021, Current Biology) more than 1644 species were included and genomes from these classes were included, except for Lichinomycetes (however there is one genome published of *Peltula cylindrica* by McDonald et al. 2013, BMC Genomics). Also entire clades of lichen-forming fungi, other than the Lichinomycetes, are missing, including within the Eurotiomycetes (i.e., Pyrenulales) and within Dothideomycetes (e.g., Trypetheliales; genome published in Haridas et al. 2020 in Studies of Mycology). Unfortunately, this biased sampling towards non lichen-forming fungi outside the Lecanoromycetes, and the absence of important classes of fungi part of the Leotiomyces is impacting the results of the ancestral state reconstruction analyses. This is greatly weakening the assumption presented throughout the manuscript that the most recent ancestor to Lecanoromycetes had to be saprotrophic and with large numbers of genes coding for PCWDEs, when this is not known. Instead of losses of genes there might be mostly gene family expansions. For example in the abstract it is written "All LFSs retain a robust PCWDE arsenal". However, the early evolution of the lecanoromycetes is on rock and other mineral substrates and a switch to other substrates is probably associated with the origin of gymnosperms and angiosperms. Under this evolutionary scenario, the ancestor to Lecanoromycetes might not have a robust PCWDE arsenal. Also, the study by Hom and Murray 2014 in Science strongly suggest a long history of fungal interactions with unicellular algae preceding the origins of lichens.

(2) One of the main conclusions presented in the abstract is that "some lichens may represent hybrid carbon assimilation systems, augmenting symbiont-derived CO₂ fixation with breakdown of lichen-external carbohydrate." However, there is strong evidence in this study suggesting that these PCWDEs could be used to attach these lichens to their organic substrates. Based on this study, there is no evidence to suggest that carbon resulting from the activity of PCWDE on substrates outside the thallus is a source of carbon for the lichen-forming fungus. This is only briefly mentioned in this study.

Other concerns and suggestions:

(3) Line 27: change "autotrophic" to "photoautotrophic"

(4) Line 28: change "phototroph fixed CO₂" to "autotroph fixed CO₂". Phototrophy is the use of

light as a source of energy to generate ATP. Autotrophy is the fixation of carbon from CO₂ into organic compounds using ATP and electrons from water.

(5) Line 79: It would be best to avoid using words like "easy".

(6) Lines 88-94: I highly recommend to remove this entire paragraph. There is no need to build a straw man that does not really exist. This is an opinion to help build a case for the importance of this study. This is not needed, the question addressed in this study is very interesting and important. As the authors wrote themselves in this paragraph "never explicitly stated, is that lichen fungal symbionts must lack the ability".

(7) Line 106-110: This statement seems flawed and weak because the main hypothesis that LFSs would exhibit functional losses in PCWDEs is based on two assumptions, one of which is that lichens evolved from saprotrophic ancestors, when there is evidence suggesting that fungi were interacting with photoautotrophs before the origin of lichens (see Hom and Murray 2014, *Science*; and Lutzoni et al. 2018, *Nature Communications*). Lichens have a very different evolution from mycorrhizal fungi. Ectomycorrhizal fungi have evolved in soils and interact with roots and we know that many agaricomycetes are saprotrophs. The evolution of the Ascomycota is very different from Basidiomycota. An hypothesis or multiples hypotheses that do not rely on these two assumptions would strengthen this study.

(8) Line 113-117: This is simply not the case. PCWDEs were not mapped across Ascomycota. Please see my point (1) above. This is a very limited and non-representative sampling across Ascomycota compare to the recent study by Li et al. April 2021 in *Current Biology*. You should focus on the second level, which is across representatives of major groups within Lecanoromycetes.

(9) Line 126: In the de.novo.sequenced column of Suppl. Table 1 you should cite the papers where existing genomes you used were published, instead of writing yes/no. Researchers that generated this genomic data will appreciate it.

(10) Line 140-158: The use of the taxon name Ascomycota throughout this paragraph (and manuscript) is misleading. Leotiomyceta is more reflective of your sampling for this study. You cannot make conclusions about Ascomycota when you sampled only a subset of the Pezizomycotina and a subset of Leotiomyceta, mainly Lecanoromycetes.

(11) Line 148-150: Suppl. Figure 11 needs to be improved. For example, I cannot see the result reported in this sentence on that figure. Lecanoromycete fungi should be delimited so that we don't have to read all the species names to determine what is what. Then, if the lichen-forming fungi could be highlighted, including Arthonia and Endocarpon, and if Stictis could be highlighted differently, this would be very helpful. The legend of this figure seems to be dissociated from the figure itself. It is not clear what (A), (B) and (C) relates to in the figure. The font size of the CAZyme abbreviations at the top of each column is way too small. You wrote "each dot represents" There are no dots on this figure.

(12) Lines 168 and 169: You wrote "among lecanoromycete genomes" and "in Lecanoromycetes", respectively. Did you intended to write Lecanoromycetidae, i.e., Lecanoromycetes excluding Ostropomycetidae? The contrast should be between Ostropomycetidae and the rest of the Lecanoromycetes.

(13) Line 174: The use of the term Lecanoromycetidae is confusing throughout the manuscript because we never know for sure if you are referring to this subclass specifically or to include Umbilicariomycetidae and Acarosporomycetidae. You should not fuse these two subclasses with Lecanoromycetidae. You should use the terms Ostropomycetidae and non-Ostro. Lecanoromycetes, for example, consistently in figures, tables, and in the text. See also lines 177, 199, 202, for example.

(14) Line 178: Change P=0 to P=0.0000

- (15) Line 195: There is no panel A for Figure 3.
- (16) Line 201: For Figure 11, it would be good to also delimit Ostropomycetidae and the OG group to better see the results reported in this paragraph.
- (17) Line 212-213: Why not also mention Thelotrema which has the highest number. Also replace Ascomycota by Leotiomyceta in this sentence.
- (18) Line 214: I could not find AA3 in Suppl. Fig. 9, and AA5 has higher numbers in the non-ostro. Lecanoromycetes than in the Ostropomycetidae.
- (19) Line 217: Change "absent across Lecanoromycetes" to "absent across Lecanoromycetes and Eurotiomycetes"
- (20) Is the cell wall composition of photobionts (e.g., Trebouxia, Trentepohlia and Nostoc) in lichens known?
- (21) Line 241: This statement starting with "In general" might be because of the sampling bias and limited sampling.
- (22) Line 245: It looks like you are referring to a different figure than Fig. 2 because this result is not shown on that figure (nothing about GH5 and GH43).
- (23) Lines 247 and 248: "occurs concomitantly with..." We don't know if LFS occurred at the base of Lecanoromycetes. It might have originated earlier. What about changing "occurs concomitantly with..." to "occurs with the origin of Lecanoromycetes, which is the largest clade of LFSs."?
- (24) Lines 246-248: This result is likely biased by the sampling. If representatives of Lichinomycetes and lichenized Eurotiomycetes (more than Endocarpon) were included you might have detected expansions rather than contractions.
- (25) Line 251: When looking at Suppl. Fig. 8, the same is true for Endocarpon and Arthonia. These important lichen-forming representatives outside Lecanoromycetes should be included more in the results and discussion of this study.
- (26) Lines 253-255: This is extremely difficult to visualize on Fig. 1. I was not sure if the grey versus dark grey of GH18 and GH43 was applying for the larger grey circles on the figures. Then I saw a minuscule light grey dot (I am not sure if real) somewhere else in the figure. Also the statement is partly incorrect because GH43 did not expand at the last ancestor of Ostropomycetidae, but rather soon after AO.
- (27) Lines 266-268: Should be written "... absence within Lecanoromycetes and other LFSs (Endocarpon and Arthonia).
- (28) Lines 274-276: This is also true for Endocarpon and Arthonia.
- (29) Line 279: "but only five lecanoromycete genomes..." What about Endocarpon and Arthonia?
- (30) Line 309: The use of the word "retain" is questionable.
- (31) Lines 314-316: But what about Endocarpon and Arthonia.
- (32) Lines 341-343: This is associated with the loss of the lichen state and should be emphasized.
- (33) Line 373: Change "the loss" by "low numbers".
- (34) Lines 373-376: One member of the OG clade is not a lichen and the other (Stictis) is often found non-lichenized. It is important to realize that species of the genus Stictis are non-lichen forming. It is clear that the Stictidaceae is an example of the lost of the lichen state. This is important. Why is that not mentioned?

(35) Lines 386-387: Change "along the lecanoromycete phylogenetic backbone" to "along the Leotiomyceta backbone".

(36) Lines 389-390: Yes, indeed!

(37) Lines 398-400: It would be good to keep in mind and consider Endocarpon and Arthonia throughout this paragraph.

(38) Lines 402-405. Another case where it would be important to consider Endocarpon and Arthonia when revising this sentence and the manuscript overall.

(39) Line 422: I would avoid the use of the term "exoskeleton" as an analogy for the cortex of lichens. Might be fine when dry, but not necessarily the case when wet.

(40) Line 430-431: Some macrolichens do not have a lower cortex.

(41) Fig. 1: The symbol in the legend for crust is different than what is used in the figure. Change "smybiotic" to "symbiotic" at the top of this column.

(42) Fig. 4: Only 72 genomes, instead of 83, are shown in Figure 4d. For that figure, it would be easier to read if the taxon names would be on the Y axis and the bar would be projecting horizontally. Also, it would be helpful to know which one are lichen forming, which one are part of the OG and of the Ostropomycetidae, for example.

REVIEWER COMMENTS

Reviewer #1 (Remarks to the Author):

Reviewer: The manuscript of Resl et al. deals with the carbohydrate degradation and transport potential in the genomes of lichen fungal symbionts. The gene inventories for plant cell wall degrading enzymes (PCWDE) across 83 ascomycete genomes including lichen fungal symbionts (LFS) and non-symbionts (29 of the genomes newly sequenced) are compared. The authors start from the hypothesis that LFS in stable associations with their phototroph symbiont would show functional losses in PCWDE, similar to what has been found for ectomycorrhizal fungi. From their comparison they find some loss in enzymes

acting on pectin, but overall all LFS retain a robust set of PCWDEs and some even have comparable gene copy numbers as well-known saprotrophic fungi, suggesting in most cases the remaining capacity to degrade cellulose and hemicellulose and to be not only dependent on the phototroph partner for carbon supply. Two predicted GH5_5 from wood inhabiting *Xylographa* species were further characterized for their cellolytic activity using heterologous expression. To test the potential use of exogenous carbohydrates by LSF via sugar transporters, the authors also compared the gene content for these transporters across all genomes and highlight the absence of Cellodextrin and maltose transporter in *Lecanoromycetes*.

Overall the data are well presented, the manuscript is very clear written and contains all necessary information and it's a very interesting topic.

Here my comments/suggestions:

My main concern comes from the lack of gene expression/transcriptomic/metatranscriptomic data (or other functional omics data like proteomics, but probably more difficult to generate). I don't understand why the authors haven't coupled the genome comparisons to "in lichen" gene expression data? The potential (presence of one or more copies of a gene) is one information. Gene regulation in the presence of the phototroph partner would be another possibility. In particular for "optimal lichen", a down-regulation of PCWDE as shown for example for the dark septate *Acephala* while forming ectomycorrhiza with pine (Miyauchi et al. 2020) could be a possibility. Or the expression of only a subset necessary for symbiosis establishment, as it has been shown for most ectomycorrhizal fungi? The authors discuss most of these possible strategies, but there remains a lot of speculation and transcriptome data might provide first clues. This information would add additional value to the findings, even if it remains descriptive.

Response: Thank you for your comments. We did indeed consider generating transcriptomic data, and our decision not to pursue this further in the context of this study is rooted in multiple considerations. Generating in vitro data under controlled conditions is technically unfeasible for most of the symbioses studied because some have not been cultured, those that have been are outstandingly slow growing (colonies <1 cm diam after 1 year) and none have been resynthesized with an alga in vitro. This leaves working with natural lichens. We have extensive experience with metatranscriptomics of natural lichens (<https://www.science.org/doi/10.1126/science.aaf8287>), but the system we worked with in that paper, *Bryoria*, we originally targeted because it forms long filaments not in contact with soil or bark, and thus we could expect the number of eukaryote species in the mix to be low(er) and manageable; we ultimately developed a workable approach to binning out transcripts and assigning them to genomes. In the early days of the present project, we tried to extend the approach from the *Bryoria* study to crust lichens such as those studied in detail in this manuscript, including a couple of the specific species we treat in this paper, but these unpublished transcriptomes were much noisier than the filament-forming *Bryorias* (for logistical reasons we also require much more starting RNA than DNA; the latter we extract from pinpoint samples using forensics protocols). The quality of these metatranscriptomes was thus too poor to use.

In theory, since we have our genomes, we could still map target mRNA to reference DNA, but the practical consequences of having high-diversity multispecies metatranscriptomes is that the crowding of species depresses sequencing coverage, and the consequence of

THAT, for the kinds of inferences you would want us to make, is that absence of any mRNA in the data set becomes meaningless — it could just be that sequencing coverage was insufficient. Of course, we could further increase coverage, but we already generate the deepest coverage shotgun data in the field, frequently dedicating 120 Gb+ of sequencing data to just 3-5 samples. In summary, we concluded that methods for generating reliable transcriptomic data for crusts that are highly integrated with their substrates need more refinement and we would have to proceed with a first level of inferences based on what we infer from presence of genes. (Also these genomes were not easy to come by: many are MAGs generated using a comparatively novel reference-free approach).

Reviewer: I further wonder why the authors focus on PCWDEs only? Carbohydrates could come from microbial cell walls (bacteria, fungi)? Any hints for this?

Response: The fact that PCWDEs featured so prominently in the abstract of the submitted version was a mis-emphasis on our part. In fact, our focus was in fact never a priori on PCWDEs; in the present version, we have replaced the abstract with a trimmed-down version of our original BioRxiv abstract, in which we used the broader wording of CAZymes. This was also our analysis from the beginning — all CAZymes first, then PCWDEs as a subset. We have adjusted the wording throughout the revised version to make this progression from broad to narrow clearer. We also add a sentence to explicitly acknowledge the potential of non-plant exogenous substrates in the Discussion.

Reviewer: P5 110 mycorrhizal fungi- true for arbuscular and ectomycorrhiza, but not for ericoid or orchid mycorrhiza

Response: We rephrased this sentence to make it clear that we are referring to arbuscular- and ectomycorrhiza.

Reviewer: P6 125 How did the authors select for the genomes to compare with? Please explain, give your criteria? This selection will somehow influence the outcome of your analysis?

Response: We have added an additional sentence to the beginning of the results (main text): “Our sampling of *Ascomycota* genomes outside of *Lecanoromycetes* was informed by two considerations: 1) selected genomes be representative of a range of CAZyme repertoires, already mapped, amongst others, by (9); and 2) it should draw primarily from *Leotiomyceta*, the group that includes the sibling classes of *Lecanoromycetes*.” We also added a sentence to supplementary discussing the sampling with respect to capturing CAZyme sets by lifestyle.

Reviewer: P9 207 This expansion of GH5_5 and GH43 in Ostropomycetidae is very interesting. Again, it would be interesting to see if they are all expressed/functional genes, in particular the sequences not close to characterized Cazymes.

Response: This is a good point and broader than just GH5_5. Our genome annotations, and in particular Saccharis analyses, highlight several hotspots of diversity of gene sequences not close to characterized CAZymes. We have added a brief section in the discussion highlighting these.

Reviewer: P10 217 So is there really any lignin-degradation potential?

Response: We appreciate that this question needed a clearer answer. We have now rewritten the section on lignin modification in the results and added a half-paragraph to the discussion outlining the pro and contra of evidence on lignin modification in lichen fungi in published literature and what evidence our survey adds.

Reviewer: P13 284 Please add here the Cazyme family (GH5_5) to make it clearer. How the two candidate genes were selected? Why two GH5_5 and not also one of the expanded GH43?

Responses:

- 1) We added the family (GH5_5) for clarity.
- 2) We added a sentence in the results to explain why these two genes were selected from the possible GH5 and GH5_5 candidates. In Supplementary Material (Section: Heterologous expression and enzymatic assays of putative LFS cellulases, page 8) we dedicated a full paragraph to how these genes were selected already in the first version.
- 3) We decided that pursuing this was beyond the scope of the present study for two reasons: 1) the main purpose of this exercise, as stated in the first sentence of this section in the results, was “To validate the functionality of putative cellulases found in lichens”, which was accomplished with the “lower-hanging fruit” of GH5_5; and 2) the substrate specificities of GH43s (where they are even known) make it very difficult to obtain substrates for enzyme assays. We agree that characterization of GH43 genes would also be interesting but would likely constitute a large-ish standalone study, the results of which would not fundamentally alter the conclusions of the present one.

Reviewer: Supplemental Table 1 Please make sure that you've the authorization for non-published genomes. Perhaps also adding the respective publications, if published? Does the ??? for the new genome accessions mean that submission is in progress?

Response: All previously unpublished genomes were produced by authors of this study. We have modified the table now to indicate original publication source for all other genomes. The question marks do indeed indicate pending submission (timed if manuscript is accepted).

Reviewer #2 (Remarks to the Author):

Reviewer: The manuscript focused on the ability of lichen fungal symbionts to degrade and transport carbohydrates based on the genome sequences. In addition to the already available genome sequences, 29 new genomes were produced to evaluate the evolution of PCWDE encoding genes. The work shows that in lecanoromycete species there are more genes encoding CAZymes and sugar transporters than previously expected.

This work builds on the previous papers of the authors, being the largest data set studied so far. Although the mechanism of carbon acquisition in lichen symbiosis is still not fully explained, the work suggests diverse models based on the comparative CAZyme arsenals in LFSs.

The work is mainly based on the comparative genomic data. I wonder why the functionality of cellulases was validated by expressing two candidate enzymes only from GH5 and none from GH43, which was also expanded CAZy family. Also, examples of recombinant cellulases from species having different life style groups would have given better insight of the functionality of LFS cellulases. To obtain better understanding how functional the expanded gene families are, wider diversity of enzymes is needed. Otherwise, the conclusions remain speculative.

Response: See also our response to Reviewer 1 on this point. Briefly, we decided to focus on validating the functionality of putative cellulases found in lichen fungi, as we felt that this is a major and lecanoromycete-wide prediction that would have been questioned had we not characterized it for at least one set of genes, and this has also not been done before. We felt we accomplished this with the “lower-hanging fruit” of GH5_5, which are closer in sequence of the putative active domains to characterized enzymes. As we note in the response to Reviewer 1, expanding to GH43 would open new technical challenges and indeed be interesting to pursue but studying this and some of the other uncharacterized CAZyme ortholog groups we found was beyond the scope of the present study.

Reviewer: The definition of lignin degrading enzymes based on the AA families should be better explained. Unfortunately, CAZy database does not provide subfamilies in AA2. Ascomycetes do not have lignin degrading peroxidases i.e LiP, MnP and VP, however genes encoding DyPs, which have been shown to degrade mono- and dimeric lignin model compounds, as well as generic peroxidases are present in ascomycete genomes. The role of laccases in lignin degradation is also controversial. I suggest to use “lignin modifying enzymes”, “lignin modification” and ability to degrade “lignin-derived compounds” to avoid misunderstanding.

Response: We agree that the first submitted version of the manuscript did not adequately explain the survey of potential lignin modification enzymes and how they relate to processes thought to operate in ascomycetes. We have made three changes: 1) we have rewritten the respective paragraph in results; 2) we have replaced the “PODs” in Figure 1 with heme haloperoxidases and DyPs from Redoxibase and made the reasons and procedure for this (and departure from CAzy) clear in the text; and 3) we added a half paragraph in the discussion briefly describing what is known especially from lichen fungi and what our survey adds to this, specifically tying to the partially characterized *LsaPOX* gene from a lichen fungus.

Reviewer: The methodology is sound with the adequate level of details and it meets the expected standards in the field. However, Figure 4 needs modifications. Figure legend should be rephrased, because Fig. 4 does not show enzyme activities, but absorbance values at 595. Also include, how many replicates have been measured to get the standard deviation. I strongly recommend to convert absorbance values to enzyme activities either in

units or katal. Y-axis shows Abs. 595 nm with values >4, which must be over the upper limit of the spectrophotometer reducing the reliability of the results. X-axes needs revision: No need to repeat pH or °C after every value, and there is a typo in pH. Please, use a symbol in β -glucan.

Response: The absorbance values in Figure 4 have been corrected for dilution factors. Absolute values were in the range of 0.1 and 1.0 absorbance values. To clear up confusion, we have changed the y axis to read “adjusted absorbance values” and added text to the figure legend. Experiments were performed in technical and biological triplicates and this information has been added to the figure legend. Unfortunately, the nature of the substrate means that it is not possible to convert absorbance values to kinetic parameters as there is no extinction coefficient for the dye.

Reviewer: In the supplementary data: Is the Figure legend 11 correct?

Response: We have improved Suppl. Figure 11 (as suggested also by Reviewer #3) and have corrected the figure legend.

Reviewer: From Figure 13 on, the text above coloured bars is not readable or only part of the text can be seen. The taxonomy bar, where different taxonomical levels are mixed, is confusing. F. ex. Most lignin modifying peroxidases and laccases (AA1_1) are grouped under Eukaryota, where also all the other presented taxonomical levels belong to. Why not to use Basidiomycota?

Response: Thank you for raising this and your suggestions to improve gene-tree plots of individual CAZyme families starting with Suppl. Fig 13. We have corrected all CAZyme family plots so that the text for different annotations is legible and we have improved the figure captions. The discrepancy in how taxonomic levels are displayed is due to the combined information of data derived from CAZy.org and our own genomes. The data gathered directly from cazy.org only distinguishes between Eukaryota, Archaea and Bacteria (see for example all experimentally characterized genes from GH12 here: http://www.cazy.org/GH12_characterized.html). On the other hand, for sequences from genomes analyzed in this study we do have more detailed taxonomic assignments. We agree that this creates a discrepancy between already characterized sequences and sequences recovered in the genomes we studied, but we do not think that resolving taxonomy for every sequence individually is necessary for the interpretation of these plots.

Reviewer: Please, check also the other figure legends. F.ex. Fig. 27, should only include genes for hemicellulose breakdown for GH10, Fig. 15 for lignin breakdown, CBMs should not include genes for lignin modification etc.

Response: We have modified the figure captions and removed ambiguity as to what is displayed in the particular figure.

Minor points:

Reviewer: Line 161 and lignin (AA1, 2 and 5).

Response: Sentence modified.

Reviewer: Line 162: Please, rephrase since lignin is not a carbohydrate.

Response: Sentence modified.

Reviewer: Lines 195 and 207: In the Figure 3 legend, Figures 3A and 3B are not mentioned.

Response: Modified Figure 3, which now has A and B.

Reviewer: Line 244-245: Please, check if referring to the Figure 2 is correct. There is no indication to GH5 and GH43 in Figure 2.

Response: Fig2. was wrongly referenced. Now this references the correct figure 3.

Reviewer: Supplementary Tables 9-11: Please, rephrase the titles beginning with “This table shows...” with brief and specific description.

Response: The titles for the tables have been rephrased.

Reviewer #3 (Remarks to the Author):

Reviewer: This is a very interesting comparative genomic study designed to determine if the number of genes coding for plant cell wall-degrading enzymes (PCWDE) are lower in lichen-forming fungi, due to their specialization on green algae or other photobionts as the potential main source of carbon. The focus of this study is on Lecanoromycetes, which is the class of fungi with the largest number of lichen-forming species. They compared 83 genomes distributed among six classes of fungi within the Leotiomyceta (Pezizomycotina, Ascomycota), with most of the genomes representing the Lecanoromycetes (46 species), Eurotiomycetes (17 species) and Sordariomycetes (15 species). Among these 83 genomes, 29 (all belonging to Lecanoromycetes) have been newly sequenced for this study. The main result is that there is a reduction in number of some gene categories coding for CAZymes, PODS and sugar transporters, however this was far from being uniform among Lecanoromycetes, many of which have high numbers of PCWDE coding-genes in the range of fungal saprotrophs. This high variation of PCWDE encoding genes among Lecanoromycetes seems to be associated with the substrate on which these fungi grow (bark, wood, soils, bryophytes, or rocks), the thallus architecture (i.e., level of contact [or attachment] of the thallus with the substrate) and if the fungus loss or partially loss the ability to form a lichen symbiotic interaction (*Stictis urceolatum*, *cyanodermella asteris*, and *Agyrium rufum*). The main result is that some lichens may be obtaining carbon not only from their photobionts but also from the breakdown of lichen-external carbohydrates.

I have two main concerns about this study: (1) the taxon sampling and the overstatement of some of the results and (2) the emphasis of the discussion and conclusion on specific aspects of the study but not others to better fit the narrative of the manuscript.

(1) Throughout the study, the authors report their results as being across Ascomycota. Unfortunately, this is far from being the case. The sampling of this study focuses on one of three subphyla of Ascomycota, i.e., Pezizomycotina. Within Pezizomycotina, they sampled six classes (mostly three classes), which are all found within the Leotiomyceta (Schoch et al. 2009, Systematic Biology). Therefore, the result of this study only applies to the Leotiomyceta, at the very most, but more specifically to the Lecanoromycetes. Even within the Leotiomyceta, no species from the Leotiomycetes (5587 species, Kirk et al. 2008), Xylonomycetes (few species), Laboulbeniomycetes (2072 species), and Lichinomycetes (350 species all lichen-forming and closely related to the Lecanoromycetes and Eurotiomycetes) were included. Yet, in the genome-scale phylogenetic study of all fungi (Li et al. April 2021, Current Biology) more than 1644 species were included and genomes from these classes were included, except for Lichinomycetes (however there is one genome published of *Peltula cylindrica* by McDonald et al. 2013, BMC Genomics). Also entire clades of lichen-forming fungi, other than the Lichinomycetes, are missing, including within the Eurotiomycetes (i.e., Pyrenulales) and within Dothideomycetes (e.g., Trypetheliales; genome published in Haridas et al. 2020 in Studies of Mycology).

Response: This echoes one of the comments by Reviewer 1 (“how did the authors select for the genomes to compare with?”). We have now added a sentence explicitly addressing our selection criteria at the beginning of the results (“Data set”). As the reviewer correctly points out, more than 1600 species were included by Li et al. and other studies such as Miyauchi also included hundreds. It is in fact precisely because such studies exist that we are able to subsample CAZyme repertoires that allow us to make statements such as we have about the size of CAZyme repertoires.

Reviewer: Unfortunately, this biased sampling towards non lichen-forming fungi outside the Lecanoromycetes, and the absence of important classes of fungi part of the Leotiomyceta is impacting the results of the ancestral state reconstruction analyses. This is greatly weakening the assumption presented throughout the manuscript that the most recent ancestor to Lecanoromycetes had to be saprotrophic and with large numbers of genes coding for PCWDEs, when this is not known.

Response: We have replace “retain” with the more neutral “possesses” wherever it occurred in the manuscript, and adjusted wording to match our conclusions in the discussion section that current sampling does not allow either hypothesis to be rejected.

Reviewer: Instead of losses of genes there might be mostly gene family expansions. For example in the abstract it is written “All LFSs retain a robust PCWDE arsenal”.

Response: That sentence is no longer in the abstract of the revised version. As to the topic of retention versus acquisition, we already discussed the range of possibilities the reviewer brings up, from retention of ancestral gene numbers to secondary expansion, in the discussion of the original submitted version (look for the sentence “Our data do not currently allow the hypothesis that the OG clade CAZyme arsenal is ancestral to be rejected, and if it is not, it becomes more likely that CAZyme loss in LFSs is driven by additional processes.”).

Reviewer: However, the early evolution of the lecanoromycetes is on rock and other mineral substrates and a switch to other substrates is probably associated with the origin of gymnosperms and angiosperms. Under this evolutionary scenario, the ancestor to Lecanoromycetes might not have a robust PCWDE arsenal. Also, the study by Hom and Murray 2014 in Science strongly suggest a long history of fungal interactions with unicellular algae preceding the origins of lichens.

Response: Thank you for your comments. This interest in gathering evidence for potential evolutionary scenarios was one major reason why we wished to explore CAZyme diversity in Lecanoromycetes, and we now have a picture of large variations within the group. As we indicate above, the saprotroph hypothesis has a long history, and some authors have reconstructed rock-dwelling ancestors as the most likely precursors of LFSs, e.g. in Eurotiomycetes. Even so, we are not aware of any data set that establishes that non-symbiotic rock oligotrophs are not, in fact, saprotrophs, and their carbon has to come from somewhere as well (nobody argues they were/are lithotrophs). Our goal in this manuscript was to map the genome-level patterns and see where the evidence is pointing, and we have provided numerous caveats about why no firm conclusion can be reached on an ancestral state already in the first version of the manuscript (discussion: unchanged in this version).

Reviewer: (2) One of the main conclusions presented in the abstract is that “some lichens may represent hybrid carbon assimilation systems, augmenting symbiont-derived CO₂ fixation with breakdown of lichen-external carbohydrate.” However, there is strong evidence in this study suggesting that these PCWDEs could be used to attach these lichens to their organic substrates.

Response: We respectfully disagree that we have provided “strong evidence” for attachment to organic substrates, but we assume that the reviewer is referring to our mention of phototroph-free parts of the mycelium including holdfasts. We are not aware that CAZymes in general have been shown to play much of a role in attachment, per se, as that role is generally taken on by adhesins; hydrolysis may facilitate cell wall remodeling but is not an ideal reaction for establishing the stable bond itself.

Reviewer: Based on this study, there is no evidence to suggest that carbon resulting from the activity of PCWDE on substrates outside the thallus is a source of carbon for the lichen-forming fungus. This is only briefly mentioned in this study.

Response: We specifically dedicate the third paragraph of the discussion to pointing readers to several lines of past experimental as well as inferential evidence. However, we do this after first spending the second paragraph of the discussion accounting for the possibility that the enzymes may be involved in algal cell wall remodeling. We are by far not the first to suggest that PCWDE-derived exogenous carbon is used as a carbon source by the lichen fungus, this is also mentioned in most recent papers by Beckett et al. and could hardly be treated as a rejectable hypothesis at this stage.

Reviewer: Other concerns and suggestions:

(3) Line 27: change “autotrophic” to “photoautotrophic”

Response: the abstract has been rewritten and this phrase no longer occurs there

Reviewer: (4) Line 28: change “phototroph fixed CO₂” to “autotroph fixed CO₂”. Phototrophy is the use of light as a source of energy to generate ATP. Autotrophy is the fixation of carbon from CO₂ into organic compounds using ATP and electrons from water.

Response: This phrase no longer occurs in the abstract.

Reviewer: (5) Line 79: It would be best to avoid using words like “easy”.

Response: We have reworded this sentence.

Reviewer: (6) Lines 88-94: I highly recommend to remove this entire paragraph. There is no need to build a straw man that does not really exist. This is an opinion to help build a case for the importance of this study. This is not needed, the question addressed in this study is very interesting and important. As the authors wrote themselves in this paragraph “never explicitly stated, is that lichen fungal symbionts must lack the ability”.

Response: We removed the paragraph.

Reviewer: (7) Line 106-110: This statement seems flawed and weak because the main hypothesis that LFSs would exhibit functional losses in PCWDEs is based on two assumptions, one of which is that lichens evolved from saprotrophic ancestors, when there is evidence suggesting that fungi were interacting with photoautotrophs before the origin of lichens (see Hom and Murray 2014, *Science*; and Lutzoni et al. 2018, *Nature Communications*). Lichens have a very different evolution from mycorrhizal fungi. Ectomycorrhizal fungi have evolved in soils and interact with roots and we know that many agaricomycetes are saprotrophs. The evolution of the Ascomycota is very different from Basidiomycota. An hypothesis or multiples hypotheses that do not rely on these two assumptions would strengthen this study.

Response: We disagree that the two assumptions we advance are flawed. It is accurate to state that lichenologists have widely assumed that symbiosis confers collective autotrophy on the lichen (we provide citations) and that hypotheses about the evolution of fungal symbionts often bring up saprotrophy. We appreciate that the reviewer does not like these assumptions, but neither of the examples they provide constitute, in our view, counter-evidence. This is because it is not settled science that lichens are defined by the mere interaction of fungi with photoautotrophs; if they were, Hom and Murray could have called their contained system a lichen (but they didn't). The fact that algal-derived polyols have hypothesized dual function (as respiratory substrate and compatible solutes) means there could be costs for respiring algal photosynthates in the lichen system, very different than in the Hom and Murray system. We already discussed this possibility briefly in the first submitted version of the manuscript (Discussion lines 381-382 in the old version, search for “respiration” in the revised version) and also have a review on this topic in press in *New Phytologist*.

Reviewer: (8) Line 113-117: This is simply not the case. PCWDEs were not mapped across Ascomycota. Please see my point (1) above. This is a very limited and non-representative sampling across Ascomycota compare to the recent study by Li et al. April 2021 in Current Biology. You should focus on the second level, which is across representatives of major groups within Lecanoromycetes.

Response: We have reworded this section.

Reviewer: (9) Line 126: In the de.novo.sequenced column of Suppl. Table 1 you should cite the papers where existing genomes you used were published, instead of writing yes/no. Researchers that generated this genomic data will appreciate it.

Response: Agreed, we have added citations.

Reviewer: (10) Line 140-158: The use of the taxon name Ascomycota throughout this paragraph (and manuscript) is misleading. Leotiomyceta is more reflective of your sampling for this study. You cannot make conclusions about Ascomycota when you sampled only a subset of the Pezizomycotina and a subset of Leotiomyceta, mainly Lecanoromycetes.

Response: We have adjusted the wording by eliminating reference to the taxon altogether where not needed, or referring to “across sampled genomes” or “other sampled Ascomycota”.

Reviewer: (11) Line 148-150: Suppl. Figure 11 needs to be improved. For example, I cannot see the result reported in this sentence on that figure. Lecanoromycete fungi should be delimited so that we don't have to read all the species names to determine what is what. Then, if the lichen-forming fungi could be highlighted, including Arthonia and Endocarpon, and if Stictis could be highlighted differently, this would be very helpful. The legend of this figure seems to be dissociated from the figure itself. It is not clear what (A), (B) and (C) relates to in the figure. The font size of the CAZyme abbreviations at the top of each column is way too small. You wrote “each dot represents” There are no dots on this figure.

Response: Thank you for the suggestions. We have corrected the Figure caption and we improved the figure with larger font-sizes and highlighted different fungal groups for better reference.

Reviewer: (12) Lines 168 and 169: You wrote “among lecanoromycete genomes” and “in Lecanoromycetes”, respectively. Did you intended to write Lecanoromycetidae, i.e., Lecanoromycetes excluding Ostropomycetidae? The contrast should be between Ostropomycetidae and the rest of the Lecanoromycetes.

Response: The text is correct as written. The first comparison is among classes of sampled Ascomycota. The next level of comparison, beginning with the following sentence (“Within Lecanoromycetes, however”) is within Lecanoromycetes.

Reviewer: (13) Line 174: The use of the term Lecanoromycetidae is confusing throughout the manuscript because we never know for sure if you are referring to this subclass specifically or to include Umbilicariomycetidae and Acarosporomycetidae. You should not

fuse these two subclasses with Lecanoromycetidae. You should use the terms Ostropomycetidae and non-Ostro. Lecanoromycetes, for example, consistently in figures, tables, and in the text. See also lines 177, 199, 202, for example.

Response: We have added a sentence where this first appears in the results to indicate that Lecanoromycetidae results are generally applicable also to Umbilicariomycetidae and Acarosporomycetidae results except where otherwise indicated, and where this is the case (especially under transporters) we specified the differences.

Reviewer: (14) Line 178: Change P=0 to P=0.0000

Response: Done

Reviewer: (15) Line 195: There is no panel A for Figure 3.

Response: This has been corrected.

Reviewer: (16) Line 201: For Figure 11, it would be good to also delimit Ostropomycetidae and the OG group to better see the results reported in this paragraph.

Response: We reworked Figure 11. This figure is much more clear now. Thank you for the suggestions.

Reviewer: (17) Line 212-213: Why not also mention Thelotrema which has the highest number. Also replace Ascomycota by Leotiomyceta in this sentence.

Response: We removed the reference to Ascomycota altogether and added Thelotrema

Reviewer: (18) Line 214: I could not find AA3 in Suppl. Fig. 9, and AA5 has higher numbers in the non-ostro. Lecanoromycetes than in the Ostropomycetidae.

Response: This paragraph has been rewritten in the revised version, and correctly linked between references to AA1 and AA2 and where they appear in the supplementary files. The reference to AA3 here was a mistake.

Reviewer: (19) Line 217: Change “absent across Lecanoromycetes” to “absent across Lecanoromycetes and Eurotiomycetes”

Response: done.

Reviewer: (20) Is the cell wall composition of photobionts (e.g., Trebouxia, Trentepohlia and Nostoc) in lichens known?

Response: Only poorly — we reviewed this work in part in FEMS Microbiology Letters (2020) and the available evidence is very limited.

Reviewer: (21) Line 241: This statement starting with “In general” might be because of the sampling bias and limited sampling.

Reviewer: (22) Line 245: It looks like you are referring to a different figure than Fig. 2 because this result is not shown on that figure (nothing about GH5 and GH43).

Response: We have corrected the reference to the figure. It now correctly refers to fig. 3.

Reviewer: (23) Lines 247 and 248: “occurs concomitantly with...” We don’t know if LFS occurred at the base of Lecanoromycetes. It might have originated earlier. What about changing “occurs concomitantly with...” to “occurs with the origin of Lecanoromycetes, which is the largest clade of LFSs.”?

Response: We have adopted the reviewer’s proposed wording.

Reviewer: (24) Lines 246-248: This result is likely biased by the sampling. If representatives of Lichinomycetes and lichenized Eurotiomycetes (more than Endocarpon) were included you might have detected expansions rather than contractions.

Response: We might or we might not in the future, but here we are simply putting words to the pattern visible on Figure 1.

Reviewer: (25) Line 251: When looking at Suppl. Fig. 8, the same is true for Endocarpon and Arthonia. These important lichen-forming representatives outside Lecanoromycetes should be included more in the results and discussion of this study.

Response: As we stated at the end of the introduction in the original version, this study is a survey of lecanoromycete genomes, representing the largest extant clade of LFSs. We have added a few sentences addressing the *Endocarpon* and *Arthonia* genomes in the revision at the position where line 402 was in the original manuscript.

Reviewer: (26) Lines 253-255: This is extremely difficult to visualize on Fig. 1. I was not sure if the grey versus dark grey of GH18 and GH43 was applying for the larger grey circles on the figures. Then I saw a minuscule light grey dot (I am not sure if real) somewhere else in the figure. Also the statement is partly incorrect because GH43 did not expand at the last ancestor of Ostropomycetidae, but rather soon after AO.

Response: We have slightly adjusted the wording about the GH43 expansion to indicate it happened just after AO. We have checked and the visual depiction of the circles is correct, however we improved the figure using colors which are easier to distinguish and we increased the size of the circles for clarity. There is also a supplementary Figure 12, which depicts the underlying CAFE results Figure 1 is based on. We reference this supplementary figure now in the caption of Figure 1.

Reviewer: (27) Lines 266-268: Should be written “.. absence within Lecanoromycetes and other LFSs (Endocarpon and Arthonia).

Response: As we stated at the end of the introduction in the original version, this study is a survey of lecanoromycete genomes. We have added a few sentences addressing the

Endocarpon and *Arthonia* genomes in the revision at the position where line 402 was in the original manuscript.

Reviewer: (28) Lines 274-276: This is also true for *Endocarpon* and *Arthonia*.

Response: As we state at the end of the introduction, this study is a survey of lecanoromycete genomes. We have added a few sentences addressing the *Endocarpon* and *Arthonia* genomes in the revision at the position where line 402 was in the original manuscript, including a reference to the absence of predicted cellobiose transporters.

Reviewer: (29) Line 279: “but only five lecanoromycete genomes...” What about *Endocarpon* and *Arthonia*?

Response: As we state at the end of the introduction and in this sentence, this study is a survey of lecanoromycete genomes. We have added a few sentences addressing the *Endocarpon* and *Arthonia* genomes in the revision at the position where line 402 was in the original manuscript.

Reviewer: (30) Line 309: The use of the word “retain” is questionable.

Response: We have replaced “retain” with “possess”.

Reviewer: (31) Lines 314-316: But what about *Endocarpon* and *Arthonia*.

Response: *Endocarpon* has *Stichococcus* and *Arthonia radiata* has *Trentepohlia* as a photobiont. We are discussing only lichens with *Trebouxia* here, to point out that fungi associated with a specific photobiont genus can have very different CAZyme profiles.

Reviewer: (32) Lines 341-343: This is associated with the loss of the lichen state and should be emphasized.

Response: This section has now been reworded to clearly state that these saprotroph origins are associated with loss of stable phototroph association.

Reviewer: (33) Line 373: Change “the loss” by “low numbers”.

Response: This is being asked as a question, and we consider it fair to ask this question. We understand that the reviewer does not favour the saprotroph hypothesis, but we discuss the evidence pro and con here and conclude this paragraph by saying that the hypothesis that this is not the case cannot be rejected (to the reviewer’s approval: comment 36).

Reviewer: (34) Lines 373-376: One member of the OG clade is not a lichen and the other (*Stictis*) is often found non-lichenized. It is important to realize that species of the genus *Stictis* are non-lichen forming. It is clear that the *Stictidaceae* is an example of the lost of the lichen state. This is important. Why is that not mentioned?

Response: The *Stictis* we used is in fact not known to have a non-lichenized state. We have inserted a sentence here to make clear that one of the sampled fungi is an endophyte, and point out in one of the next sentences that the clade includes 500 non-LFSs, mostly saprotrophs.

Reviewer: (35) Lines 386-387: Change “along the lecanoromycete phylogenetic backbone” to “along the Leotiomyceta backbone”.

Response: We understand that the reviewer wants greater certainty for the whole backbone, but we intentionally wished to specifically highlight the uncertainty of the lecanoromycete backbone in this case.

Reviewer: (36) Lines 389-390: Yes, indeed!

Reviewer: (37) Lines 398-400: It would be good to keep in mind and consider *Endocarpon* and *Arthonia* throughout this paragraph.

Response: We have added a few sentences addressing the *Endocarpon* and *Arthonia* genomes in the revision at the position where line 402 was in the original manuscript.

Reviewer: (38) Lines 402-405. Another case where it would be important to consider *Endocarpon* and *Arthonia* when revising this sentence and the manuscript overall.

Response: We have added a few sentences addressing the *Endocarpon* and *Arthonia* genomes in the revision at the position where line 402 was in the original manuscript.

Reviewer: (39) Line 422: I would avoid the use of the term “exoskeleton” as an analogy for the cortex of lichens. Might be fine when dry, but not necessarily the case when wet.

Response: We have replaced “exoskeleton” with “scaffold”, perhaps a more neutral term, which has already been used elsewhere for the cortex.

Reviewer: (40) Line 430-431: Some macrolichens do not have a lower cortex.

Response: Valid point, this has been reworded to say “the cortex mediates the passage of environmental molecules for many macrolichens”

Reviewer: (41) Fig. 1: The symbol in the legend for crust is different than what is used in the figure. Change “smybiotic” to “symbiotic” at the top of this column.

Response: done

Reviewer: (42) Fig. 4: Only 72 genomes, instead of 83, are shown in Figure 4d. For that figure, it would be easier to read if the taxon names would be on the Y axis and the bar would be projecting horizontally. Also, it would be helpful to know which one are lichen forming, which one are part of the OG and of the *Ostropomycetidae*, for example.

Response: The figure caption has been modified to indicate that these are the 72 genomes in which orthologs were found.

Reviewers' Comments:

Reviewer #1:

Remarks to the Author:

This is the second time that I received the manuscript entitled "Large differences in carbohydrate degradation and transport potential in the genomes of lichen fungal symbionts" by Resl et al. for review.

I still think that the addition of gene expression data would have been very useful and would have prevented some speculation, but I understand and accept the difficulties and technical issues founded in the lichen biology (and not so promptly resolvable).

All my other comments/questions have been taken into account and mistakes have been corrected, so that I'm satisfied with the current version of the manuscript.

Reviewer #2:

Remarks to the Author:

All my concerns about the manuscript have been well addressed.

Reviewer #3:

Remarks to the Author:

(1) As part of my second review of this manuscript, I would like to comment on the response from the authors to Reviewer #1. The first response from the authors to Reviewer # 1 states that "Generating in vitro data under controlled conditions is technically unfeasible for most of the symbioses studied because some have not been cultured, those that have been are outstandingly slow growing (colonies < 1 cm diam after 1 year) and none have been resynthesized with an alga in vitro." The latter part of this statement is not accurate. Transcriptomic data has been generated by Armaleo et al. (2019) for *Cladonia grayi* growing axenically, its photobiont *Asterochloris* growing axenically, and both growing together in co-culture, i.e., early stages of symbiotic resynthesis.

(2) I agree with reviewer #1 that transcriptomic data is very important. I also understand that comparative transcriptomic studies are onerous. At the very least, the authors should take advantage of the comparative transcriptomic study done by Armaleo et al. (2019) for a typical lichen (*Cladonia*, *Lecanoromycetes*). I believe the authors might find answers to some of the questions they are addressing here for specific genes. This is an excerpt from the abstract of Armaleo et al. (2019): "In coculture, the fungus upregulated small secreted proteins, membrane transport proteins, signal transduction components, extracellular hydrolases and, notably, a ribitol transporter and an ammonium transporter,". The results of the study by Armaleo et al. (2019) need to be fully explored and integrated in this study.

Armaleo et al. 2019. The lichen symbiosis re-viewed through the genomes of *Cladonia grayi* and its algal partner *Asterochloris glomerata*. *BMC Genomics* 20:605.

(3) The revised version of this manuscript and the response to my first comment (1) is not addressing the poor sampling of this study for *Leotiomyceta*. The authors responded: "This echoes one of the comments by Reviewer 1 ("how did the authors select for the genomes to compare with?"). We have now added a sentence explicitly addressing our selection criteria at the beginning of the results ("Data set"). As the reviewer correctly points out, more than 1600 species were included by Li et al. and other studies such as Miyauchi also included hundreds. It is in fact precisely because such studies exist that we are able to subsample CAZyme repertoires that allow us to make statements such as we have about the size of CAZyme repertoires."

However, that sentence is too vague and does not explain why so many genomes were excluded.

(4) No response was provided to my comment #21.

(5) Line 27: Is the word "stabilized" the appropriate term for a nutritional symbiosis?

(6) Lines 97 and 98: You wrote "and historical assumptions that they evolved from saprotrophic ancestors," this is according to whom? You are stating here that lichens evolved from saprotrophic ancestors, but you refer to three papers on mycorrhizae (9,23,24) at the end of this sentence. There is no foundation in the literature supporting the premise to your hypothesis "that LFSs would exhibit functional losses in CAZymes coinciding with the beginning of stable association with phototroph symbionts". If there are "historical assumptions that [lichens] evolved from saprotrophic ancestors" please cite these papers. Otherwise, your premise does not stand.

Lines 120-122: You wrote "Because the few published lecanoromycete genomes are not representative of deep evolutionary splits in the group, we generated 29 new genomes for this study". First, *Lasallia* and *Umbilicaria* metagenomic data are from a deep split within *Lecanoromycetes* as shown on your Figure 1. These genomes were taken from the literature for this study. The metagenome of *Acarospora strigata*, which represents the deepest or second deepest split within *Lecanoromycetes* was also sequenced a while back by McDonald et al. (2013). All these genomes are available publicly. Second, except for *Hypocenomyce* and new genomic data for *Acarospora*, the selected taxa are not representing the deepest lecanoromycete lineages. You need to modify this sentence to reflect the current reality.

Lines 145, 151, 164, 165, 195, 322: Replace "Ascomycota" by "Leotiomyceta".

Responses to Reviewer's Comments (Resl et al., NCOMMS-21-35456A)

Reviewer #1 (Remarks to the Author)

This is the second time that I received the manuscript entitled "Large differences in carbohydrate degradation and transport potential in the genomes of lichen fungal symbionts" by Resl et al. for review.

I still think that the addition of gene expression data would have been very useful and would have prevented some speculation, but I understand and accept the difficulties and technical issues founded in the lichen biology (and not so promptly resolvable).

All my other comments/questions have been taken into account and mistakes have been corrected, so that I'm satisfied with the current version of the manuscript.

Response: Thank you for your comments. We agree on the desirability of gene expression data and hope this problem can be resolved in the future.

Reviewer #2 (Remarks to the Author)

All my concerns about the manuscript have been well addressed.

Response: Thank you.

Reviewer #3 (Remarks to the Author)

(1) As part of my second review of this manuscript, I would like to comment on the response from the authors to Reviewer #1. The first response from the authors to Reviewer # 1 states that "Generating in vitro data under controlled conditions is technically unfeasible for most of the symbioses studied because some have not been cultured, those that have been are outstandingly slow growing (colonies < 1 cm diam after 1 year) and none have been resynthesized with an alga in vitro." The latter part of this statement is not accurate. Transcriptomic data has been generated by Armaleo et al. (2019) for *Cladonia grayi* growing axenically, its photobiont *Asterochloris* growing axenically, and both growing together in co-culture, i.e., early stages of symbiotic resynthesis.

Response: We acknowledge that the co-cultures of *C. grayi* have generated fungal-algal bundling, which is considered by many authors to constitute "resynthesis", though fully natural thalli are not formed.

(2) I agree with reviewer #1 that transcriptomic data is very important. I also understand that comparative transcriptomic studies are onerous. At the very least, the authors should take advantage of the comparative transcriptomic study done by Armaleo et al. (2019) for a typical lichen (*Cladonia*, *Lecanoromycetes*). I believe the authors might find answers to some of the questions they are addressing here for specific genes. This is an excerpt from the abstract of Armaleo et al. (2019): "In coculture, the fungus upregulated small secreted proteins, membrane transport proteins, signal transduction components, extracellular hydrolases and, notably, a ribitol transporter and an ammonium transporter,". The results of the study by Armaleo et al. (2019) need to be fully explored and integrated in this study.

Armaleo et al. 2019. The lichen symbiosis re-viewed through the genomes of *Cladonia grayi* and its algal partner *Asterochloris glomerata*. *BMC Genomics* 20:605.

Response: We have incorporated a reference to two supplementary file sheets from Armaleo et al. (2019) into the current revision; however, we note that fungal CAZymes and MFS transporters of the type we analyze in detail in our study are not addressed in detail in the main text of the Armaleo et al. study.

(3) The revised version of this manuscript and the response to my first comment (1) is not addressing the poor sampling of this study for Leotiomyceta. The authors responded: “This echoes one of the comments by Reviewer 1 (“how did the authors select for the genomes to compare with?”). We have now added a sentence explicitly addressing our selection criteria at the beginning of the results (“Data set”). As the reviewer correctly points out, more than 1600 species were included by Li et al. and other studies such as Miyauchi also included hundreds. It is in fact precisely because such studies exist that we are able to subsample CAZyme repertoires that allow us to make statements such as we have about the size of CAZyme repertoires.”

However, that sentence is too vague and does not explain why so many genomes were excluded.

Response: None of the patterns we report in this manuscript require us to use hundreds of more genomes, and as such these were not excluded per se from any analysis that otherwise would have required them. We acknowledged in a previous round that the sample could at most place limitations on the interpretation of CAZyme gene expansion and contraction patterns in ancestral state reconstruction, but already in the first round of revisions we adjusted our wording in regards to such statements to be very conservative.

(4) No response was provided to my comment #21.

Response: Your comment (21) was: “Line 241: This statement starting with “In general” might be because of the sampling bias and limited sampling.” The relevant wording no longer occurred in the revised version; apologies that we neglected to explicitly note that.

(5) Line 27: Is the word “stabilized” the appropriate term for a nutritional symbiosis?

Response: The term “stability” is frequently used in the context of symbioses. As we have argued elsewhere, including in the back-and-forth responses to this reviewer’s comments, we do not consider it settled science that lichen symbioses (plural) are all nutritional symbioses — nutrition may play a role in some, but the evidence is surprisingly ambiguous.

(6) Lines 97 and 98: You wrote “and historical assumptions that they evolved from saprotrophic ancestors,” this is according to whom? You are stating here that lichens evolved from saprotrophic ancestors, but you refer to three papers on mycorrhizae (9,23,24) at the end of this sentence. There is no foundation in the literature supporting the premise to your hypothesis “that LFSs would exhibit functional losses in CAZymes coinciding with the beginning of stable association with phototroph symbionts”. If there are “historical assumptions that [lichens] evolved from saprotrophic ancestors” please cite these papers. Otherwise, your premise does not stand.

Response: This is a fair point — we have both slightly tweaked the wording and added a citation here (the new #23) of a recent review in which we looked at, amongst other things,

reconstructed ancestral states for the antecedents of lichen fungi, as well as historical assumptions on what they must have descended from. There exist multiple such studies and the easier approach is to cite the review here.

Lines 120-122: You wrote “Because the few published lecanoromycete genomes are not representative of deep evolutionary splits in the group, we generated 29 new genomes for this study”. First, Lasallia and Umbilicaria metagenomic data are from a deep split within Lecanoromycetes as shown on your Figure 1. These genomes were taken from the literature for this study. The metagenome of Acarospora strigata, which represents the deepest or second deepest split within Lecanoromycetes was also sequenced a while back by McDonald et al. (2013). All these genomes are available publicly. Second, except for Hypocenomyce and new genomic data for Acarospora, the selected taxa are not representing the deepest lecanoromycete lineages. You need to modify this sentence to reflect the current reality.

Response: We have reworded this sentence to better reflect the overall strategy we applied to sampling, which is also discussed in greater detail in the methods.

Lines 145, 151, 164, 165, 195, 322: Replace “Ascomycota” by “Leotiomyceata”.

Response: After the first round of reviews, we inserted, upon previous request from this reviewer, two mentions of the unranked taxon ‘Leotiomyceata’ at the beginning of the results section, and thenceforth referred to “sampled Ascomycota”, which we feel acknowledges the narrowed nature of the taxon sampling. We prefer not to over-use the term ‘Leotiomyceata’ as it is not actually a valid taxon under the Code of Nomenclature, we are not convinced its emphasis will always be helpful for understanding Ascomycota evolution, and its repeated use after being mentioned once is in our view not imperative.